# A Closer Look at Weakly-Supervised Audio-Visual Source Localization

**Shentong Mo**
Carnegie Mellon University

**Pedro Morgado**
University of Wisconsin-Madison

## Abstract

Audio-visual source localization is a challenging task that aims to predict the location of visual sound sources in a video. Since collecting ground-truth annotations of sounding objects can be costly, a plethora of weakly-supervised localization methods that can learn from datasets with no bounding-box annotations have been proposed in recent years, by leveraging the natural co-occurrence of audio and visual signals. Despite significant interest, popular evaluation protocols have two major flaws. First, they allow for the use of a fully annotated dataset to perform early stopping, thus significantly increasing the annotation effort required for training. Second, current evaluation metrics assume the presence of sound sources at all times. This is of course an unrealistic assumption, and thus better metrics are necessary to capture the model's performance on (negative) samples with no visible sound sources. To accomplish this, we extend the test set of popular benchmarks, Flickr SoundNet and VGG-Sound Sources, in order to include negative samples, and measure performance using metrics that balance localization accuracy and recall. Using the new protocol, we conducted an extensive evaluation of prior methods, and found that most prior works are not capable of identifying negatives and suffer from significant overfitting problems (rely heavily on early stopping for best results). We also propose a new approach for visual sound source localization that addresses both these problems. In particular, we found that, through extreme visual dropout and the use of momentum encoders, the proposed approach combats overfitting effectively, and establishes a new state-of-the-art performance on both Flickr SoundNet and VGG-Sound Source. Code and pre-trained models are available at https://github.com/stoneMo/SLAVC.

## 1 Introduction

Humans and most other animals have evolved to localize sources of sound in their environment. This remarkable ability relies in part on the uniqueness of different sound sources, which allows us to recognize the sounds we hear and visually localize them in our environment. Given recent advances in audio and visual perception research, there is broad interest in developing multi-modal systems capable of mimicking our ability to visually localize sound sources.

One promising direction is to leverage the co-occurrence between sounds and the corresponding sources in video data. Since audio-visual co-occurrence arises naturally, algorithms can scale to very large datasets without requiring costly human annotations. However, despite encouraging recent progress [1, 2, 3, 4, 5], currently accepted evaluation protocols hide two critical limitations of current methods: (1) current methods overfit easily even when scaled up to large datasets, and (2) current methods assume that visible sound sources are always present in the video and thus are unreliable when deployed on realistic data where this assumption does not hold.

The first limitation remained hidden as prior works [2, 4, 6, 5] rely heavily on early stopping for optimal performance (*i.e.*, by continuously validating the model during training using a human-

36th Conference on Neural Information Processing Systems (NeurIPS 2022).

annotated set). However, this practice violates the main assumption of weakly supervised visual source localization, *i.e.*, the requirement for no bounding box annotations during training.

The second limitation remained hidden as most prominent benchmark datasets [1, 6] and evaluation metrics only assess the ability to localize sound sources when one is present in the video. Importantly, it ignores the ability to correctly predict the *absence* of visual sound sources. This has led to a bias towards localization accuracy with disregard for false positive detection. False positive detection is closely related to the silent object detection problem highlighted in recent works [7, 8]. However, although these works introduce methods to suppress the localization of silent objects, they require the collection of a clean dataset containing a single source per video. Instead, we aim to tackle this issue without assuming knowledge of the number of sources.

The two limitations above highlight the need for a more balanced and complete evaluation protocol for visual sound source localization. To achieve this, we extend popular benchmark test sets (Flickr SoundNet [1] and VGG-Sound Sources [6]) to include 'negative' samples without any visible sound sources. We conduct an extensive evaluation of existing methods [2, 3, 4, 6, 9, 7], where in addition to overall localization accuracy, we also assess methods based on their ability to predict negative samples. We observe that all prior work suffers from significant overfitting problems, relying on early stopping for optimal performance. We also found that previous approaches struggle to strike a good balance between false positive and false negative rates.

We also propose a novel procedure - Simultaneous Localization and Audio-Visual Correspondence (`SLAVC`) - which provably tackles these two issues. First, to combat overfitting, we adopt slow-moving momentum target encoders and extreme visual dropout. Second, to reduce false positives, we localize visual sources by forcing the model to *explicitly* perform audio visual correspondence in addition to localization. The latter term highlights which regions within an image are most associated with a particular audio, while the former downplays regions that can be better described by the audio of other samples. By combining these two terms, the model is able to both accurately localize visible sources and identify when no visible sources are present. Using the newly developed evaluation protocol, we show that, unlike all prior work, `SLAVC` does not overfit and thus can be trained without relying on early stopping. `SLAVC` achieves state-of-the-art performance on multiple datasets and is more accurate at identifying samples without any visible sound sources.

## 2 Related Work

**Audio-Visual Self-Supervised Learning.** The natural audio-visual alignment is a rich source of supervision for self-supervised learning [10]. Recently, it has been explored to learn a wide variety of deep learning models [11, 12, 13, 1, 14, 15, 16, 17, 18, 19]. Given a database of videos, the main idea is to close the distance between audio and visual features from the same video while pushing away those from different videos [20, 19, 21], from the same video but different timestamp [22, 23], or from the same video but difference spatial location [24]. Such audio-visual alignment is beneficial to several tasks, such as audio separation [18, 25, 26, 15, 16, 27, 28, 29], audio-visual spatialization [14, 30, 31, 24], visual sound source localization [1, 2, 3, 4, 6, 32]. In this work, we mainly focus on visual source localization, which requires learning fine-grained and high-resolution representations that are discriminative of the various sound sources.

**Audio-Visual Source Localization.** Audio-visual source localization aims at predicting the location of sounding objects in a video. Early models [33, 34, 35] learn to capture the low-level correspondences between audio and visual features. Recently, contrastive approaches [1, 23, 2, 6, 9] seek to localize objects by aligning audio and visual representation spaces. For instance, LVS [6] leveraged a contrastive loss with hard negative mining to learn the audio-visual co-occurrence map discriminatively. Contrastive learning with hard positives was introduced in HardPos [9] to learn audio-visual alignment with negative samples. EZ-VSL [5] introduced a multiple instance contrastive learning framework that focus only on the most aligned regions when matching the audio to the video.

However, we show that these methods can easily overfit in the source localization task, and heavily rely on early stopping for best performance. Furthermore, with the exception of [7, 8], they focus on localizing sound sources that are visible, and struggle to identify negatives (when there are no visible sources). Although understanding when sources are not visible is important to several audio-visual tasks, like in open-domain audio-visual source separation [36, 37], this issue still poses significant challenges. DSOL [7] and IEr [8] proposed a method to suppress localization of silent objects, and

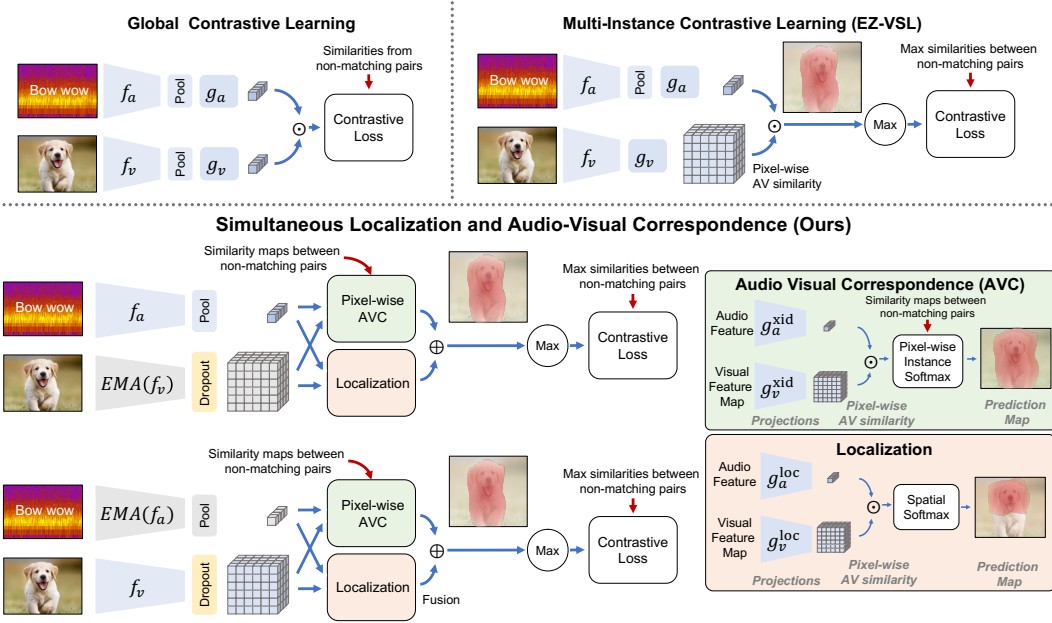

Figure 1: Illustration of the proposed `SLAVC` with synchronized momentum audio-visual matching. The audio-visual correspondence of pixel-wise similarity between momentum branches and base branches is aligned by the cross-modal multiple instance contrastive learning objective.

an evaluation metric that measures the ability to ignore them. These methods, however, rely on the knowledge of the number of sound sources, not available in most large scale datasets. In this work, we do not assume such knowledge. We propose a novel Simultaneous Instance Discrimination and Localization framework for weakly-supervised visual source localization with clear benefits, as it demonstrably improves both localization accuracy and false positive rates. We also annotate real world test samples that may lead to false positives (beyond silent objects), and propose new metrics for a more comprehensive evaluation.

**Weakly-supervised Object Detection (WSOD).** Weakly-supervised object detection [38, 39, 40, 41] is closely related to visual sound localization, since both aim at localizing object regions by learning from image/frame level supervision (corresponding audio in the case of VSL, and object classes for WSOD). However, different from WSOD, we assume no access to class information during training. Instead, `SLAVC` aims at learning from audio-visual correspondence alone.

## 3 Visual Source Localization

We propose a self-supervised learning approach for Visual Sound Localization (VSL). This method builds on the current state-of-the-art based on multiple-instance contrastive learning [5]. The proposed approach addresses two critical problems: 1) severe overfitting even when trained on large datasets and 2) the tendency to hallucinate sound sources when none are visible in the video. For clear exposition, we begin by defining the problem and briefly revisit the current state-of-the-art [5] (Sec. 3.1). We then detail the proposed approach in Sec. 3.2, and highlight the main differences to prior work. Finally, we present in 4 an evaluation protocol that is sensitive to these issues.

### 3.1 Preliminaries: Weakly-Supervised Visual Source Localization

Given an audio-visual dataset $\mathcal{D} = \{(v_i, a_i) : i = 1, \ldots, N\}$, VSL seeks to train a system that can predict the location of the sources present in sound $a_i$ within the visual frame $v_i$. VSL models can be learned from various levels of supervision: *Unsupervised VSL* learns to localize without any form of human annotations; *Weakly-supervised VSL* forgo bounding-box supervision but may leverage categorical information. This categorical information can be in the form of object labels for visual encoder pre-training, or audio event labels for audio encoder pre-training; *Semi-supervised*

*VSL* learns from a small number of bounding-box annotations, together with a large unsupervised or weakly-supervised dataset; Finally, *Fully supervised VSL* models rely on large amounts of fully annotated datasets. In this paper, we tackle the Weakly-Supervised VSL problem. We note that equivalent prior work, such as [1, 4, 2, 3, 7, 6, 9, 5], often refer to this problem as unsupervised VSL, despite actually addressing the weakly supervised problem, as they use vision models pre-trained on ImageNet for object recognition.

To train the system, the current state-of-the-art method, EZ-VSL [5] maps both audio and visual features into a common space using projection heads $g_a(\cdot)$ and $g_v(\cdot)$, resulting in a set of visual features spanning all locations in an image $V_i = \{g_v(\mathbf{v}_i^{xy}) : \forall x, y\}$ and one global audio feature $g_a(\mathbf{a}_i)$. Ideally, the model should be trained to align the audio and visual features at locations $(x, y)$, where sound sources are located. However, since these locations are unknown during training, EZ-VSL [5] optimizes a multiple-instance contrastive learning objective, in which the audio representation $g_a(\mathbf{a}_i)$ is matched with *at least one* location in the corresponding bag of visual features $V_i$, while pushing away from the visual features in all "negative" bags in the same mini-batch $V_j \; \forall j \neq i$ (at all spatial locations). Specifically, the model is trained to minimize the average per sample loss

$$\mathcal{L}_i^{\text{MICL}} = -\log \frac{\exp\left(\frac{1}{\tau}\texttt{sim}(\mathbf{a}_i, V_i)\right)}{\sum_{j=1}^{B} \exp\left(\frac{1}{\tau}\texttt{sim}(\mathbf{a}_i, V_j)\right)} - \log \frac{\exp\left(\frac{1}{\tau}\texttt{sim}(\mathbf{a}_i, V_i)\right)}{\sum_{k=1}^{B} \exp\left(\frac{1}{\tau}\texttt{sim}(\mathbf{a}_k, V_i)\right)} \tag{1}$$

where $\tau$ is a temperature hyper-parameter, $B$ is the batch-size, and the similarity $\texttt{sim}(\mathbf{a}_i, V_j)$ between an audio feature $\mathbf{a}_i$ and a bag of visual features $V_j$ is computed by max-pooling audio-visual cosine similarities $s()$ across spatial locations $x, y$

$$\texttt{sim}(\mathbf{a}_i, V_j) = \max_{xy} s(g_a(\mathbf{a}_i), g_v(\mathbf{v}_j^{xy})). \tag{2}$$

### 3.2 Simultaneous Localization and Audio-Visual Correspondence (SLAVC)

Previous methods, including EZ-VSL [5], suffer from two critical limitations: (1) models easily overfit to the self-supervised objective; (2) models are not capable of identifying negative samples (*i.e.*, samples with no visible sound sources). The method proposed in this work is designed to address these two limitations. To combat overfitting, we apply two regularization techniques: dropout [42] and (slow-moving) momentum encoders [43, 44]. To better identify negatives, we propose to conduct VSL by simultaneously performing sounding region localization and audio-visual correspondence. These three components are illustrated in Figure 1, and are now discussed separately.

**Dropout** [42] is a widely used technique to combat overfitting. We applied dropout on the output of both the $f_v$ and $f_a$ encoders. Interestingly, we found that heavy visual dropout is required to prevent overfitting (with a dropout probability as large as 0.9), while audio dropout is not required. Since multiple-instance contrastive learning (Eq. 1) requires the audio to match only one of a large number of locations in the image ($h \times w$), spurious alignment becomes more likely. Visual dropout reduces the likelihood of spurious alignments, as visual features become more spatially redundant.[1]

**Momentum encoders** [43, 44] is a technique often used in self-supervised and semi-supervised learning to obtain slow-moving target representations, leading to more stable self-training and enhanced representations. We apply momentum encoders to both audio and visual inputs $\hat{\mathbf{a}}_i = \hat{f}_a(a_i), \hat{\mathbf{v}}_i = \hat{f}_v(v_i)$, in order to obtain more stable targets. Following [43, 44], momentum encoders are updated using an exponential moving average of the corresponding online encoders with coefficient $m$.[2] Since updates to the audio-visual encoders are slowly incorporated into the momentum encoders, the target representations display a smoother behavior during the training process.

**Simultaneous localization and audio-visual correspondence** To better avoid false predictions, we *explicitly* force the model to only predict regions of an image that can be used to identify the corresponding sound source. Specifically, we factorize VSL into two terms. The first is a localization term $P^{\text{loc}}$ that distinguishes regions within an image that likely depict the sound source from the regions that likely do not. The second is an audio-visual correspondence term $P^{\text{avc}}$ that highlights

---

[1]Spatial redundancy is encouraged as it prevents loss of information when features are dropped.

[2]EMA update is $\hat{\theta} \leftarrow \hat{\theta} + (1-m)\theta$, where $\theta$ and $\hat{\theta}$ denote the parameters of online and momentum functions.

regions that are likely associated with the corresponding audio, and suppresses regions that would be better explained by other sound sources. The key insight is that to prevent false detections, VSL should select only the regions that are *simultaneously* highlighted in both terms. While the first term is responsible for sound source localization, the second prevents VSL to be overly confident when no sound sources are visible in the image.

Localization and audio-visual correspondence is conducted on separate subspaces. In particular, let $g_a^{\mathrm{loc}}(\mathbf{a}_i)$ and $g_a^{\mathrm{avc}}(\mathbf{a}_i)$ be the two audio representations of $a_i$, and $\{g_v^{\mathrm{loc}}(\mathbf{v}_i^{xy}) : \forall x, y\}$ and $\{g_v^{\mathrm{avc}}(\mathbf{v}_i^{xy}) : \forall x, y\}$ the two bags of visual features for $v_i$. We use linear projections for the various functions $g(\cdot)$. Then, we define the localization term as the softmax $\rho_{xy}(\cdot)$ over spatial dimensions $x, y$

$$P^{\mathrm{loc}}(\mathbf{a}_i, \mathbf{v}_j^{xy}) = \rho_{xy}\left(\frac{1}{\tau} s\left(g_a^{\mathrm{loc}}(\mathbf{a}_i), g_v^{\mathrm{loc}}(\mathbf{v}_j^{xy})\right)\right), \tag{3}$$

and the audio-visual correspondence term as the softmax $\rho_i(\cdot)$ over instances $i$

$$P^{\mathrm{avc}}(\mathbf{a}_i, \mathbf{v}_j^{xy}) = \rho_i\left(\frac{1}{\tau} s(g_a^{\mathrm{avc}}(\mathbf{a}_i), g_v^{\mathrm{avc}}(\mathbf{v}_j^{xy}))\right). \tag{4}$$

To select the regions in which both $P^{\mathrm{loc}}$ and $P^{\mathrm{avc}}$ are active, the two terms are combined into a single prediction map $P(\mathbf{a}_i, \mathbf{v}_j^{xy}) = P^{\mathrm{loc}}(\mathbf{a}_i, \mathbf{v}_j^{xy}) \cdot P^{\mathrm{avc}}(\mathbf{a}_i, \mathbf{v}_j^{xy})$, and the model is trained to optimize

$$\mathcal{L}_i^{\mathrm{SLAVC}} = -\log \frac{\max_{xy} P(\mathbf{a}_i, \mathbf{v}_i^{xy})}{\sum_{j=1}^B \max_{xy} P(\mathbf{a}_i, \mathbf{v}_j^{xy})} - \log \frac{\max_{xy} P(\mathbf{a}_i, \mathbf{v}_i^{xy})}{\sum_{k=1}^B \max_{xy} P(\mathbf{a}_k, \mathbf{v}_i^{xy})}. \tag{5}$$

**Full method**  While we demonstrate the benefits of the three components separately, they can be combined for optimal performance. The full approach is illustrated in Fig. 1. Dropout is always applied on the encoders' outputs. To combine the SLAVC factorization with momentum encoders, the model is trained to optimize

$$\mathcal{L}_i^{\mathrm{full}} = -\log \frac{\max_{xy} P(\mathbf{a}_i, \hat{\mathbf{v}}_i^{xy})}{\sum_{j=1}^B \max_{xy} P(\mathbf{a}_i, \hat{\mathbf{v}}_j^{xy})} - \log \frac{\max_{xy} P(\hat{\mathbf{a}}_i, \mathbf{v}_i^{xy})}{\sum_{k=1}^B \max_{xy} P(\hat{\mathbf{a}}_k, \mathbf{v}_i^{xy})}, \tag{6}$$

where $\hat{\mathbf{a}}_i$ and $\hat{\mathbf{v}}_j^{xy}$ are the audio and visual momentum features.

During inference, both localization and audio-visual correspondence terms are used for VSL. We observed that matching momentum features at test time lead to improved localization. Thus, given an audio-visual pair $(v, a)$, the audio-visual localization map at location $xy$ is computed as

$$\mathbf{S}_{\mathrm{AVL}}^{xy} = s\left(g_a^{\mathrm{loc}}(\hat{\mathbf{a}}), g_v^{\mathrm{loc}}(\hat{\mathbf{v}}^{xy})\right) + s\left(g_a^{\mathrm{avc}}(\hat{\mathbf{a}}), g_v^{\mathrm{avc}}(\hat{\mathbf{v}}^{xy})\right). \tag{7}$$

Furthermore, this audio-visual localization map can be easily combined with object-guided localization proposed in [5] for improved performance.

## 4  Benchmarking Visual Source Localization

We introduce an evaluation protocol for VSL that is more sensitive to the high false positives and overfitting issues of current approaches.

First, to ensure overfitting is not hidden by the evaluation protocol, we suggest to ***rule out early stopping*** from weakly-supervised VSL evaluation, and instead always evaluate models after training them to convergence (or a large number of iterations). Note that early stopping defeats the purpose of weakly-supervised VSL, as it requires a fully annotated evaluation subset for tracking performance.

Second, to assess false detection of non-existing sources, we extend the test sets of both Flickr-SoundNet [1] and VGG Sound Sources [45] to include samples without visible sound sources. We also use metrics that measure the balance between high localization accuracy and low false positive rates.

**Extended Flickr-SoundNet/VGG-SS**  Non-visible sounds or frames with silent objects are prevalent in video. To tackle these cases, we present a new evaluation protocol. We extended VGG-SS/Flickr-SoundNet by merging clips with no sounding objects to the original test sets. Specifically,

Table 1: Statistics of weakly-supervised audio-visual source localization test sets.

| Dataset | Small | Medium | Large | Huge | Total Pos | Real Neg | Automated Easy Neg | Automated Hard Neg | Total Neg | Total |
|---|---|---|---|---|---|---|---|---|---|---|
| Ground-truth size (pixels) | $1-32^2$ | $32^2-96^2$ | $96^2-144^2$ | $144^2-224^2$ | $1-224^2$ | 0 | 0 | 0 | 0 | $1-224^2$ |
| Flickr-SoundNet [1] | 0 | 9 | 83 | 158 | 250 | 0 | 0 | 0 | 0 | 250 |
| Extended Flickr-SoundNet | 0 | 9 | 83 | 158 | 250 | 42 | 169 | 39 | **250** | **500** |
| VGG-Sound Source [6] | 134 | 1796 | 1726 | 1502 | 5158 | 0 | 0 | 0 | 0 | 5158 |
| Extended VGG-SS | 134 | 1796 | 1726 | 1502 | 5158 | 379 | 3594 | 1185 | **5158** | **10316** |

we analyzed 1000 videos from VGG-Sound test set (and 250 from Flickr-SoundNet test set), and manually select 5-second clips with non-audible frames and/or non-visible sound sources. This resulted in 379/42 samples with no sounding sources for VGG-SS/Flickr-SoundNet, respectively.

Beyond the manually identified negative pairs, we further generate negative samples by pairing audio and videos that do not belong together. We control the difficulty of these negative pairs, by sampling 25% of pairs from audio and video that are from the same class (hard negative), and 75% from different classes. We merge all negatives with the VGG-SS [6] and Flickr-SoundNet [1] test set. Table 1 shows the statistics of the extended test sets. We also split test samples into four groups (Small, Medium, Large, and Huge) according to the size of the sound sources, as measured by the area (in pixels) occupied by the ground-truth bounding boxes.

**Evaluation metrics** Localization maps are often evaluated by comparing them to a group of human annotations using consensus intersection over union (cIoU, denoted as $u$) [1]. Given a set of predictions with cIoUs $\mathcal{U} = \{u_i\}_{i=1}^N$, prior work [2, 3, 4, 6, 9, 5] measures the localization accuracy (LocAcc) among all samples with visible sounding objects, where each prediction is considered to be correct if its cIoU is above the cIoU threshold $\gamma^3$. The cIoU threshold is set at $\gamma = 0.5$ unless otherwise specified.

Beyond localization error, we also evaluate on samples with no visible sounding sources. Thus, the model cannot assume the presence of a sound source, it is required to predict whether the current video contains a visible source or not. This is accomplished by computing a confidence score $d_i$, which we define as the maximum value in the predicted audio-visual similarity map, $\max_{xy} S_{\text{AVL}}^{xy}$. For evaluation, we define a flag $c_i$ to be 1 for samples with visible sources (positive samples) and 0 for samples with no visible sources (negative samples). True positive are then given as $\mathcal{TP} = \{i|c_i = 1, d_i > \delta, u_i > \gamma\}$, false positives as $\mathcal{FP} = \{i|c_i = 1, d_i > \delta, u_i \leq \gamma\} \cup \{i|c_i = 0, d_i > \delta\}$, and false negatives as $\mathcal{FN} = \{i|c_i = 1, d_i \leq \delta\}$. These sets are used to compute the Average Precision (AP), and the maximum F1 (max-F1) score obtained by sweeping the confidence threshold (*i.e.* $\max_\delta F1(\delta)$). Detailed formulas are provided in appendix.

# 5 Experiments

Using the evaluation protocol outlined above, we now show that prior works easily overfit and are prone to high false positive rates. We also show that the proposed SLAVC effectively addresses these issues, outperforming prior state-of-the-art by large margins. Next, we conduct a thorough analysis of the various components of the system, and identify the main limitations of both ours and prior work, namely, small objects and high-quality detection.

## 5.1 Experimental setup

**Datasets** We evaluate the effectiveness of the proposed method on two datasets - Flickr SoundNet [1] and VGG Sound Sources [45]. Following commonly-used settings [6, 9, 5], we use a subset of 144k samples for training in both cases. We evaluate on the extended test sets described in Sec. 4.

**Models and optimization** We followed prior work and used ResNet-18 [46] for both the audio and visual encoders. The visual encoder is initialized with ImageNet [47] pre-trained weights [6, 9, 5]. The output dimensions of the audio and visual encoders (*i.e.*, the output of projection functions $g()$) was kept at 512, the momentum encoders update factor at 0.999, and the visual dropout at 0.9. No audio dropout is applied. Models are trained with a batch size of 128 on 2 GPUs for 20 epochs (which

---

[3]This metric is also referred to as "CIoU". To avoid confusion, we prefer the term Localization Accuracy.

Table 2: Comparison results of LocAcc ("CIoU") for models obtained with and without early stopping on Flickr SoundNet and VGG-SS testsets. All models were trained on VGG-Sound 144k. ⋆ indicates values reported in the original papers.

| Method | Flickr-SoundNet | | VGG-SS | |
| | Early Stop | NO Early Stop | Early Stop | NO Early Stop |
|---|---|---|---|---|
| Attention10k [1] | 42.26 | 34.16 | 18.50⋆/18.52 | 14.04 |
| CoarsetoFine [4] | – | 47.20 | – | 21.93 |
| DMC [2] | 55.60 | 52.80 | 23.90 | 22.63 |
| AVObject [3] | – | – | 29.70⋆ | – |
| DSOL [7] | 74.00 | 72.91 | 29.91 | 26.87 |
| LVS [6] | 71.90⋆/71.60 | 19.60 | 34.40⋆/33.36 | 10.43 |
| HardPos [9] | 76.80⋆ | – | 34.60⋆ | – |
| EZ-VSL [5] | 79.60 | 66.40 | 34.28 | 31.58 |
| SLAVC (ours) | **83.20** | **83.60** | **37.22** | **37.79** |
| EZ-VSL + OGL [5] | 83.94⋆/83.94 | 72.80 | 38.85⋆/38.85 | 37.86 |
| SLAVC (ours) + OGL [5] | **86.40** | **86.00** | **39.67** | **39.80** |

Table 3: Comparison results of the proposed metrics (AP, max-F1, LocAcc) on Extended Flickr-SoundNet and Extended VGG-SS benchmark. All models were trained on VGG-Sound 144k.

| Method | Extended Flickr-SoundNet | | | Extended VGG-SS | | |
| | AP | max-F1 | LocAcc | AP | max-F1 | LocAcc |
|---|---|---|---|---|---|---|
| Center Prior | – | – | 67.60 | – | – | 34.16 |
| CoarsetoFine [4] | 0.00 | 38.20 | 47.20 | 0.00 | 19.80 | 21.93 |
| LVS [6] | 9.80 | 17.90 | 19.60 | 5.15 | 9.90 | 10.43 |
| Attention10k [1] | 15.98 | 24.00 | 34.16 | 6.70 | 13.10 | 14.04 |
| DMC [2] | 25.56 | 41.80 | 52.80 | 11.53 | 20.30 | 22.63 |
| DSOL [7] | 38.32 | 49.40 | 72.91 | 16.84 | 25.60 | 26.87 |
| OGL [5] | 40.20 | 55.70 | 77.20 | 18.73 | 30.90 | 36.58 |
| EZ-VSL [5] | 46.30 | 54.60 | 66.40 | 24.55 | 30.90 | 31.58 |
| SLAVC (ours) | **51.63** | **59.10** | **83.60** | **32.95** | **40.00** | **37.79** |
| EZ-VSL + OGL [5] | 48.75 | 56.80 | 72.80 | 27.71 | 34.60 | 37.86 |
| SLAVC (ours) + OGL [5] | **52.15** | **60.10** | **86.00** | **34.46** | **41.50** | **39.80** |

we found to be enough to achieve convergence in most cases). We used the Adam [48] optimizer with $\beta_1 = 0.9$, $\beta_2 = 0.999$, learning rate of $1e-4$ and weight decay of $1e-4$. Our implementation, available at https://github.com/stoneMo/SLAVC, is based on PyTorch [49] deep learning tool.

**Audio and visual processing** We extract audio-visual pairs composed of a single frame and 3 seconds of audio centered around the frame. The visual frame is resized into 256 along the shortest edge, followed by random cropping and random horizontal flipping. During inference, frames are resized into 224×224, with no additional data augmentations. For the audio, we extract log spectrograms with 257 frequency bands over 300 timesteps from 3s of audio at 22050 kHz. The underlying short-term Fourier transform is computed on approximately 25ms windows with a step size of 10ms.

**Prior works and baselines** We compare the proposed approach to several prior VSL methods. Specifically, we considered Attention 10k [1], CoarsetoFine [4], DMC [2], DSOL [7], LVS [6], HardPose [9] and EZ-VSL [5]. We used authors' implementations when available, and our own implementations otherwise. To ensure that our results match the original papers, we report both results when possible in Tab. 2. For the current state-of-the-art EZ-VSL [5], we consider two versions: with and without object guided localization (OGL). OGL computes an object prior which is merged with the audio-visual localization prediction for improved localization. We also use OGL together with our approach. As a sanity check, we also evaluate a Center Prior baseline, which always selects a circle centered in the middle of the image. Since when recording, we tend to put objects in the middle of the frame, this baseline provides a good low bound for localization performance, which VSL methods should outperform.

| vdrop | AP | max-F1 | LocAcc |
|---|---|---|---|
| 0 | 26.03 | 32.00 | 36.45 |
| 0.50 | 26.33 | 32.20 | 36.27 |
| 0.75 | 30.22 | 37.80 | 36.82 |
| 0.90 | **32.95** | **40.00** | **37.79** |
| 0.95 | 26.07 | 33.70 | 35.94 |

(a) Video dropout.

| adrop | AP | max-F1 | LocAcc |
|---|---|---|---|
| 0 | **32.95** | **40.00** | **37.79** |
| 0.50 | 20.63 | 28.30 | 32.30 |
| 0.75 | 18.88 | 24.70 | 26.91 |
| 0.90 | 16.21 | 22.70 | 19.10 |
| 0.95 | 7.63 | 12.60 | 10.64 |

(b) Audio dropout.

| mom | AP | max-F1 | LocAcc |
|---|---|---|---|
| 0. | 25.03 | 32.50 | 32.45 |
| 0.9 | 27.12 | 33.50 | 34.32 |
| 0.99 | 25.58 | 32.00 | 35.40 |
| 0.995 | 30.53 | 37.40 | 35.94 |
| 0.999 | **32.95** | **40.00** | **37.79** |

(c) Momentum.

| SLAVC Train | Infer | AP | max-F1 | LocAcc |
|---|---|---|---|---|
|  | AVLoc | 28.19 | 35.60 | 32.75 |
| ✓ | AVLoc | 22.01 | 30.80 | 23.65 |
| ✓ | AVC | 17.29 | 29.60 | 34.72 |
| ✓ | SLAVC | **32.95** | **40.00** | **37.79** |

(d) SLAVC decomposition.

Table 4: **Ablation studies.** Impact of video dropout rate (vdrop), audio dropout rate (adrop), the momentum parameter for the target encoders (mom), and the use of SLAVC decomposition both during training and inference. Default parameters are highlighted in gray.

## 5.2 Main results

**Preventing overfitting** To demonstrate that current methods suffer from severe overfitting, we compare several methods trained with and without early stopping [1, 2, 3, 4, 6, 9, 5]. Table 2 shows the Localization Accuracy (LocAcc) of these models on two datasets: Flickr SoundNet and VGG Sound Sources. We observe that, despite the large training sets (144k audio-visual pairs in both datasets), early stopping is critical to obtain high LocAcc in all prior works, despite the fact that it should not be used in a weakly-supervised VSL setting. In fact, we observed that in most cases performance peaks within the first 2 to 3 training epochs, and decays rapidly afterwards. Training curves are shown in Figure **??**. This observation suggests that, due to overfitting, prior methods do not scale well (*i.e.*, they cannot take advantage of larger datasets).

In contrast, SLAVC does not show the same signs of overfitting. The model improves as it is trained for longer. As a result, early stopping is not required to obtain a high performing model. Also, since the model can better leverage the large training data, it also significantly outperforms all prior work (even without early stopping). Without OGL (object-guided localization), we outperform the previous SOTA (EZ-VSL [5]) by 3.60% and 2.94% on Flickr and VGG-SS, respectively. If early stopping is ruled out, these gains increase to 17.2% and 6.2% on Flickr and VGG-SS, respectively. Finally, by combining SLAVC with OGL, we achieve a new state-of-the-art on weakly-supervised VSL (86.4% LocAcc on Flickr and 39.67% on VGG-SS).

**Preventing false positives** Since all prior work rely on LocAcc as the main evaluation metric, models are not penalized for high false positives rates. To better assess prior work, we evaluated several models on the proposed Extended Flickr and VGG-SS datasets with the evaluation protocol described in Sec. 4 (without early stopping). As can be seen in Table 3, with the exception of EZ-VSL and SLAVC, AP and max-F1 scores of prior works are very low, as these models struggle to avoid false positives without substantially increasing false negatives. EZ-VSL was shown to be more effective at preventing false positives. However, its overall performance was still lacking in comparison to the proposed SLAVC, which achieves the best results on all metrics. SLAVC+OGL improves AP by 3.4 and max-F1 by 3.3 on the Extended Flickr-SoundNet test set. On the more challenging extended VGG-SS dataset, the gains of SLAVC+OGL were even larger, with an increase in AP of 6.75 and in max-F1 score of 6.9%.

## 5.3 Analysis

We now conduct a thorough analysis of SLAVC. Our goal is to improve audio-visual localization of sound sources. As seen in Tables 2 and Tab. 3, OGL enhances localization. However, since it only provides an object prior, OGL can be applied to any method. To better understand the proposed procedure, we conduct the various analysis without OGL (unless stated otherwise).

**Regularization strategies.** We start by showing in Tables 4a, 4b and 4c the effect of the various regularization techniques used to tackle overfitting (dropout and momentum target encoders). Models are trained and tested on VGG Sound Sources. As can be seen, heavy dropout of visual features (*i.e.* with a dropout rate of 0.9) is critical for localization, while audio dropout is not required. As for the target encoders, we found that a momentum of 0.999 achieves best results, outperforming models that do not use momentum encoders by significant margins (6.0% AP, 5.2% max-F1, and 2.0% LocAcc).

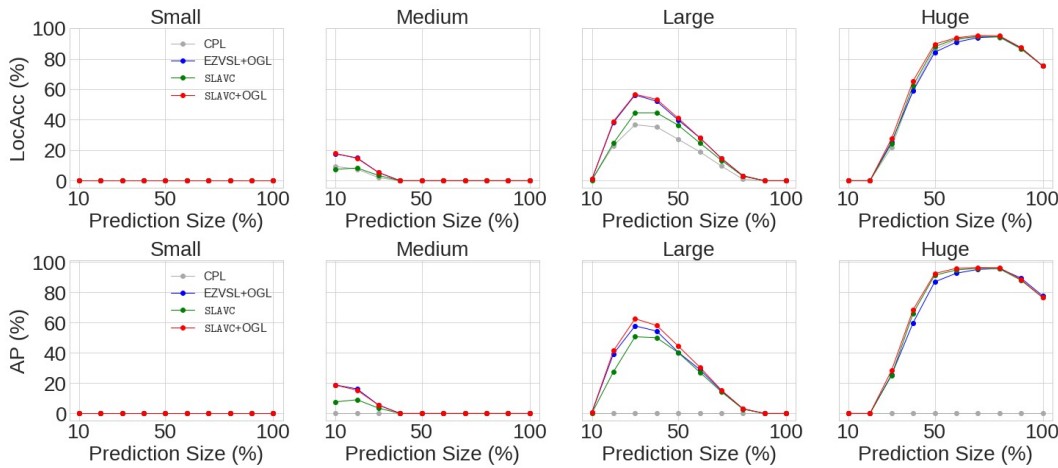

Figure 2: Effect of different relative sorting thresholds on LocAcc and AP for visible objects of various sizes (Small, Medium, Large, Huge). CPL, EZVSL+OGL, SLAVC, and SLAVC+OGL denote the localization using Gaussian center prior map, audio-visual response map, object-guided map, and the linear combination of audio-visual response and object-guided map.

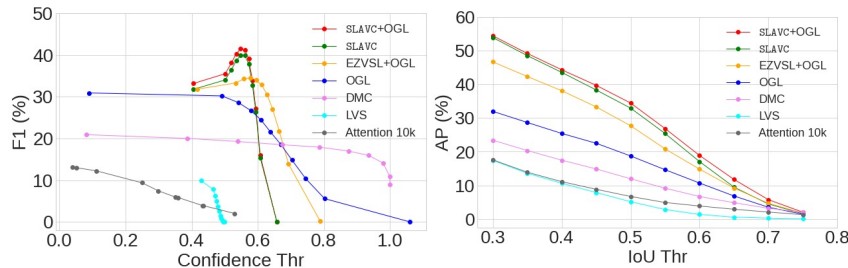

Figure 3: Effect of confidence thresholds on the F1 score and IoU thresholds on AP.

Table 5: Comparison results of max-F1 score for false positives among hand selected negatives, easy negatives and hard negatives on Extended Flickr-SoundNet and Extended VGG-SS benchmark where models are trained on VGG-Sound 144k data.

| Method | Extended Flickr-SoundNet | | | Extended VGG-SS | | |
| --- | --- | --- | --- | --- | --- | --- |
| | Real Neg | Automated Easy Neg | Automated Hard Neg | Real Neg | Automated Easy Neg | Automated Hard Neg |
| LVS [6] | 14.50 | 17.80 | 14.30 | 8.30 | 8.80 | 7.60 |
| Attention10k [1] | 27.10 | 28.10 | 27.00 | 9.10 | 7.40 | 7.90 |
| CoarsetoFine [4] | 36.90 | 39.40 | 35.80 | 22.10 | 19.80 | 18.90 |
| DMC [2] | 48.80 | 41.40 | 43.70 | 20.70 | 19.40 | 21.90 |
| EZ-VSL [5] | 52.60 | 55.70 | 54.20 | 32.80 | 36.10 | 31.40 |
| SLAVC (ours) | **63.50** | **57.40** | **63.30** | **40.30** | **43.20** | **33.00** |

**SLAVC decomposition.** We also studied the effect of SLAVC decomposition during training and the localization strategy used for inference, *i.e.*, either using audio-visual similarities tuned for localization alone (AVLoc; $s(\hat{\mathbf{v}}^{\text{loc}}_{xy}, \hat{\mathbf{a}}^{\text{loc}})$), for AV correspondence (AVC; $s(\hat{\mathbf{v}}^{\text{avc}}_{xy}, \hat{\mathbf{a}}^{\text{avc}})$) or both (SLAVC; Eq. 7). Table 4d shows the localization performance for the different strategies. We observe that by training to simultaneously perform both audio-visual localization and correspondence, we can enhance localization by significant margins (4.76% AP, 4.40% max-F1, and 5.04% LocAcc). Also, both SLAVC branches, audio-visual localization (AVLoc) and correspondence (AVC), are required during inference for optimal performance.

**Sound source size.** Figure 2 shows the performance of Center Prior (CPL), OGL, `SLAVC` and `SLAVC`+OGL on sources of different sizes (*i.e.* using the Small, Medium, Large and Huge subsets) on the VGG Sound Sources datasets. Each method was forced to output a prediction of constant size, by setting an appropriate threshold in the localization map. We then plotted the LocAcc obtained for predictions that vary between 10% and 90% of the total image area. As can be seen, localization performance degrades significantly for smaller objects, regardless of the method used. In fact, all methods achieve a peak LocAcc of 0 in the Small subset containing sources of size up to $32^2$ in area. This corroborates observations in prior work, that detection of small sound sources remains one of the main limitations of current approaches. We also observe that all methods perform well when localizing very large sound sources. In fact, since the IoU threshold is set relatively low ($\gamma = 0.5$) given the prediction sizes, even the Center Prior, which always selects the center pixels, can achieve high LocAcc. The methods differ however when they are required to also identify negative samples, in which case `SLAVC` significantly outperforms prior methods (as shown in AP scores) regardless of prediction size.

**Balancing positive and negative detection.** To better understand the models' ability to detect samples with no visible sound sources, we plot in Fig. 3 (left) the F1 score for increasing confidence thresholds. As the threshold increases, models are more selective in identifying sound sources, and thus avoid making predictions if none is visible. However, this can come at the cost of missing visible sound sources. As can be seen in Fig. 3 (left), `SLAVC` and EZVSL are the only approaches whose F1 scores increase with higher confidence thresholds, with `SLAVC` achieving the highest F1 score. This shows that unlike most of prior work, `SLAVC` can effectively identify negative pairs, *i.e.* ti reduces false positive rates without sacrificing true positive rates.

**High quality detection.** Figure 3 (right) also shows the AP scores at varying IoU thresholds. We observe that while our method consistently outperforms prior work, the performance of all methods degrade substantially as the IoU threshold increases. In fact, the AP at $0.75$ IoU is close to 0 for all methods, indicating that current weakly-supervised VSL methods still struggle to achieve high quality detection.

**Negative type.** Table 5 studies the max-F1 score for false positives among real and hand selected negatives (easy/hard). For each of the three negative subsets, we add a similar number of positives (e.g., there are 1185 hard negatives on Extended VGG-SS, so 1185 random positives are added to this set to get a total of 2370 samples). These results measure how well methods can balance positive and negative detections when given only those types of negatives. The proposed `SLAVC` achieves the best performance in terms of all types of negatives, which further demonstrates the robustness and effectiveness in balancing positive and negative detections.

## 6 Conclusion

In this work, we identify two critical issues with current weakly-supervised visual sound source localization methods: severe overfitting even when trained with large datasets, and their poor ability to identify when no sound sources are visible (*i.e.*, negatives). Since current evaluation protocols allow for early stopping and always assume the presence of a visible sound sources, they are not sensitive to the aforementioned issues, reason why they remained relatively unknown. To fix these issues, we propose a new evaluation protocol and a novel method for VSL. We extend current evaluation datasets to also include negative samples (*i.e.*, frames with no visible sound source). We show that overfitting can be effectively addressed through commonly used regularization techniques like dropout and momentum target encoders. We also show that forcing the model to explicitly conduct both localization and audio visual correspondence enhances the model's ability to identify negative samples.

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
