# A Closer Look at Weakly-Supervised Audio-Visual Source Localization
## *(Supplementary Material)*

**Shentong Mo**
Carnegie Mellon University

**Pedro Morgado**
University of Wisconsin-Madison

In this appendix, we detail the baselines and metrics used for benchmarking visual sound localization (LocAcc, F1 score, and Average Precision). We then demonstrate the effectiveness of the proposed SLAVC in other commonly used training datasets (Flickr 10k) and other settings like semi-supervised localization and open set visual sound source localization. Finally, we compare the qualitative results of our approach with existing methods.

Code is available at: `https://github.com/stoneMo/SLAVC`.

## 1 Baselines

We conducted a comprehensive benchmarking study of existing approaches. For a fair comparison, we use the same backbone-ResNet18 [1] for all baselines. Namely, we considered:

- Attention 10k [2] (2018'CVPR): the first attention-based work with a two-stream architecture with each modality for weakly-supervised sound source localization in an image, and this approach was extended to semi-supervised settings with ground-truth maps in 5k Flickr set; (code: `https://github.com/ardasnck/learning_to_localize_sound_source`)

- DMC [3] (2019'CVPR): a multi-modal clustering network for learning audiovisual correspondences by using convolutional maps with each modality in different shared spaces; (code: `https://github.com/DTaoo/Simplified_DMC`, MIT License)

- CoarsetoFine [4] (2020'ECCV): a two-stage pipeline that aligned the cross-modal features in a coarse-to-fine way; (code: `https://github.com/shvdiwnkozbw/Multi-Source-Sound-Localization`)

- LVS [5] (2021'CVPR): a contrastive learning framework with hard negatives mining to extract the audio-visual co-occurrence map discriminatively; (code: `https://github.com/hche11/Localizing-Visual-Sounds-the-Hard-Way`, Apache License 2.0)

- HardPos [6] (2022'ICASSP): an improved work based on LVS by adding hard positives for aligning audio-visual matching semantics from negative pairs;

- EZ-VSL [7] (2022'ECCV): a strong baseline that proposed the multiple instance contrastive learning to align locations with high similarity in the image and push away from all locations in different images; (code: `https://github.com/stoneMo/EZ-VSL`, Apache License 2.0)

- DSOL [8] (2020'NeurIPS): a two-stage training baseline to deal with silence in category-aware sound source localization; (code: `https://github.com/DTaoo/Discriminative-Sounding-Objects-Localization`, MIT License);

## 2 Localization Accuracy, F1 Score, Average Precision

Consider a set of samples $\mathcal{D} = \{(v_i, a_i) : i = 1, \ldots, N\}$, ground-truth maps $\mathcal{G} = \{\mathcal{G}_i\}_{i=1}^N$ and predicted maps $\mathcal{S} = \{\mathcal{S}_i\}_{i=1}^N$. In map based localization, each prediction $\mathcal{S}_i$ is obtained by first computing pixel-wise localization scores $S_i \in \mathbb{R}^{H \times W}$, and then applying a thresold $\alpha$, $\mathcal{S}_i(\alpha) = \{(x,y)|S_i(x,y) > \alpha\}$. To evaluate each prediction $\mathcal{S}_i$, prior work [2] relies on intersection

over union between $\mathcal{S}_i$ and $\mathcal{G}_i$. IoU is calculated as

$$IoU_i(\alpha) = \frac{\sum_{xy \in \mathcal{S}_i(\alpha)} g_{xy}}{\sum_{xy \in \mathcal{S}_i(\alpha)} g_{xy} + \sum_{xy \in \mathcal{S}_i(\alpha) - \mathcal{G}_i} 1} \tag{1}$$

where $\mathcal{G} = \{(x,y)|g_{xy} > 0\}$ denotes the the set of pixels that represent sounding objects, and $g_{xy} \in [0,1]$ the ground-truth evidence that a sounding objects lies at location $(x,y)$. Since in some datasets ground-truth is collected from multiple annotators, consensus IoU (cIoU) is used instead. Refer to [2] for details on how multiple annotations are merged to compute $g_{xy}$.

Since we're interested in assessing the model's performance both when sounding objects are present or not, we allow the model to make no predictions. To do this, in addition to the localization prediction map $\mathcal{S}_i$, the model is asked to output a confidence score $c_i$. If the confidence score $c_i$ is too low (below a threshold $\delta$), the model predicts an empty set (*i.e.*, the original $\mathcal{S}_i$ is considered invalid). Predictions $\mathcal{S}_i$ are considered correct if $IoU_i$ are above a pre-specified threshold $\gamma$. Under these definitions, true positives, false positives and false negatives are computed as

$$\mathcal{TP}(\gamma,\delta) = \{i|\mathcal{G}_i \neq \emptyset, IoU_i > \gamma, c_i > \delta\} \tag{2}$$
$$\mathcal{FP}(\gamma,\delta) = \{i|\mathcal{G}_i \neq \emptyset, IoU_i \leq \gamma, c_i > \delta\} \cup \{i|\mathcal{G}_i = \emptyset, c_i > \delta\} \tag{3}$$
$$\mathcal{FN}(\gamma,\delta) = \{i|\mathcal{G}_i \neq \emptyset, c_i \leq \delta\} \tag{4}$$

**Localization Accuracy** To evaluate the localization accuracy of our models among samples with visible sound sources, we measure the localization accuracy at a predefined *IoU* threshold $\gamma$

$$\text{LocAcc}(\gamma,\delta) = \frac{|\mathcal{TP}(\gamma,\delta)|}{|\{i|\mathcal{G}_i \neq \emptyset\}|}. \tag{5}$$

Since this metric only measures localization performance, it assumes that all samples contain a visible source to be localized. We thus ignore the confidence threshold (*i.e.*, $\delta = -\infty$), so as to predict a source location for every sample. Most existing work [3, 9, 4, 5, 6, 7] report LocAcc at $\gamma = 0.5$ under the name of "CIoU". We use this metric when comparing to results reported in the original papers.

While the localization accuracy provides a good metric to evaluate how accurate predictions are for samples with sounding objects, it does not assess how accurately models can ignore samples with NO sounding objects. Thus, to comprehensively evaluate both sounding and non-sounding samples, we also evaluate our models using F1 score and average precision (AP).

**F1 score** balances precision and recall,

$$\text{F1}(\gamma,\delta) = \frac{2 * \text{Precision}(\gamma,\delta) * \text{Recall}(\gamma,\delta)}{\text{Precision}(\gamma,\delta) + \text{Recall}(\gamma,\delta)}, \tag{6}$$

where

$$\text{Precision}(\gamma,\delta) = \frac{|\mathcal{TP}(\gamma,\delta)|}{|\mathcal{TP}(\gamma,\delta)| + |\mathcal{FP}(\gamma,\delta)|} \quad \text{and} \quad \text{Recall}(\gamma,\delta) = \frac{|\mathcal{TP}(\gamma,\delta)|}{|\mathcal{TP}(\gamma,\delta)| + |\mathcal{FN}(\gamma,\delta)|}. \tag{7}$$

However, $\text{F1}(\gamma,\delta)$ depends on how strict the confidence threshold $\delta$ is set (the highest $\delta$ is set, the more samples are predicted as non-sounding). To find the optimal balance, we compute $\text{F1}(\gamma,\delta)$ for all values of $\delta$ and report the max-F1 score

$$\text{max-F1}(\gamma) = \max_{\delta} \text{F1}(\gamma,\delta). \tag{8}$$

**Average precision** (AP) is another metric often used in object detection. To compute AP, we closely follow [10]. The only difference is that, when computing the Precision-Recall curve, we do not perform 11 point interpolation. We compute the full curve (without interpolation). Refer to [10] for details on AP computation.

## 3 Training curves

Prior work, including previous state-of-the-art EZ-VSL [7] overfit to the self-supervised loss. To better see this, we plotted the training curve of both EZ-VSL and our method SLAVC in Fig. 1. We plot both LocAcc and AP obtained on the extended VGG-SS test set as the model is trained on the VGG-SS 144k. As can be seen, the localization performance of EZ-VSL peaks very early (around epoch 6), and degrades significantly afterwards. We also observe that EZ-VSL has a more unstable training behavior than SLAVC, which completely avoids overfitting and thus continuously improves localization performance as it is trained.

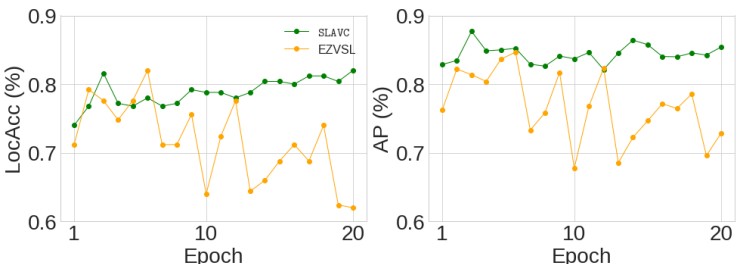

Figure 1: Training curves on VGG-SS 144k: training epoch vs **LocAcc** and training epoch vs **AP** for both the current state-of-the-art EZ-VSL [7] and the proposed SLAVC.

Table 1: Comparison results of weakly-supervised and semi-supervised training on Flickr SoundNet testset where models are trained on Flickr-10k data. $\star$ indicates values reported in the original papers.

| Method | Early Stop | | | NO Early Stop | | |
|---|---|---|---|---|---|---|
| | AP | max-F1 | LocAcc | AP | max-F1 | LocAcc |
| *weakly-supervised:* | | | | | | |
| Attention10k [2] | 45.02 | 49.90 | 43.60$\star$/42.54 | 20.28 | 32.80 | 19.60 |
| DMC [3] | 53.17 | 70.80 | 54.80 | 51.92 | 70.50 | 54.40 |
| LVS [5] | 68.92 | 71.80 | 58.20$\star$/59.20 | 8.46 | 15.50 | 8.40 |
| EZ-VSL [7] | 75.64 | 76.20 | 62.65 | 70.54 | 73.10 | 57.60 |
| SLAVC (w/o OGL) | **87.10** | **90.10** | **82.00** | **87.10** | **90.10** | **82.00** |
| EZ-VSL + OGL [7] | 84.56 | 89.40 | 81.93$\star$/81.93 | 83.60 | 89.90 | 81.60 |
| SLAVC (ours) + OGL [7] | **88.45** | **91.80** | **84.80** | **88.45** | **91.80** | **84.80** |
| *semi-supervised:* | | | | | | |
| Attention10k [2] | 82.75 | 88.28 | 82.80$\star$/82.70 | – | – | – |
| SLAVC (ours) | **85.94** | **91.10** | **83.60** | – | – | – |
| SLAVC (our) + OGL [7] | **88.01** | **92.50** | **86.00** | – | – | – |

## 4 Weakly/Semi Supervised Results on Flickr-10k

In addition to the weakly-supervised setting, some works also explore the use of a small annotated dataset to guide training [2]. To better understand the added value of a small number of bounding boxes, we also train our model in the semi-supervised setting. Following [2], we use the ground-truth maps of 5000 Flickr images to directly supervise SLAVC's predictions through the additional loss

$$\mathcal{L}_{semi} = \frac{1}{\mathbf{1}_i^{sup}} \sum_{i=1}^{B} \mathbf{1}_i^{sup} \|S_{\text{AVL}}(v_i, a_i) - G_i\|^2, \qquad (9)$$

where $S_{\text{AVL}}$ denote the model's prediction of **??**, $\mathbf{1}_i^{sup}$ indicates whether sample $i$ has ground-truth annotations, $G_i$ represents the ground-truth localization map, and $\|\cdot\|$ represents the norm over all spatial locations $x, y$.

To compare the semi-supervised to the weakly-supervised setting, we train the model on the Flickr-10k [2] (containing 10k samples from Flickr). Results are reported in Table 1. We don't report results with the latest checkpoint in the semi-supervised case, since the available ground-truth annotations can be used for early stopping. Nevertheless, our SLAVC achieves the state-of-the-art results compared to existing methods in both settings. The additional annotations allow our model to achieve higher LocAcc. However, the difference between the two settings (weakly and semi supervised) is much smaller in our case, when compared to Attention10k. This result indicates that the gains achieved by Attentio10k with the additional supervision were mostly due to the overfitting and false positive issues identified in this work.

## 5 Open Set Results

To evaluate the generalization of our model beyond sound sources heard during training, we follow previous work [5, 7] and train the model on 70k data with 110 heard categories in VGG-Sound dataset [11]. Table 2 reports the comparison results on heard and unheard 110 classes. We achieve significant improvements against previous methods. For instance, using models without OGL and

Table 2: Comparison results on VGG-SS for open set audio-visual localization trained on 70k data with heard 110 classes. * indicates values reported in the original papers.

| Test class | Method | Early Stop | | | NO Early Stop | | |
|---|---|---|---|---|---|---|---|
| | | AP | max-F1 | LocAcc | AP | max-F1 | LocAcc |
| Heard 110 | Attention10k [2] | – | – | – | 11.03 | 27.20 | 18.78 |
| | CoarsetoFine [4] | – | – | – | 0.00 | 33.50 | 20.09 |
| | DMC [3] | 22.35 | 35.60 | 21.68 | 23.13 | 36.30 | 22.15 |
| | LVS [5] | 28.67 | 43.00 | 28.90*/28.48 | 20.10 | 33.90 | 20.40 |
| | EZ-VSL [7] (w/o OGL) | 33.95 | 49.00 | 32.49 | 32.80 | 46.90 | 30.62 |
| | EZ-VSL [7] (w OGL) | 36.48 | 53.30 | 37.25*/36.35 | 36.81 | 53.90 | 36.93 |
| | SLAVC (w/o OGL) | **38.08** | **52.40** | **35.53** | **38.51** | **52.80** | **35.84** |
| | SLAVC (w OGL) | **40.38** | **55.00** | **37.95** | **40.84** | **55.30** | **38.22** |
| Unheard 110 | Attention10k [2] | – | – | – | 15.72 | 27.30 | 15.91 |
| | CoarsetoFine [4] | – | – | – | 0.00 | 38.20 | 23.57 |
| | DMC [3] | 24.24 | 39.00 | 24.23 | 24.69 | 39.50 | 24.62 |
| | LVS [5] | 26.04 | 41.00 | 26.30*/26.01 | 19.42 | 32.80 | 19.65 |
| | EZ-VSL [7] (w/o OGL) | 32.87 | 49.50 | 32.93 | 33.63 | 45.60 | 29.55 |
| | EZ-VSL [7] (w OGL) | 38.19 | 55.50 | 39.57*/38.37 | 38.04 | 55.30 | 38.21 |
| | SLAVC (w/o OGL) | **36.97** | **53.30** | **36.35** | **37.27** | **53.50** | **36.50** |
| | SLAVC (w OGL) | **39.24** | **56.00** | **38.87** | **39.19** | **55.90** | **38.83** |

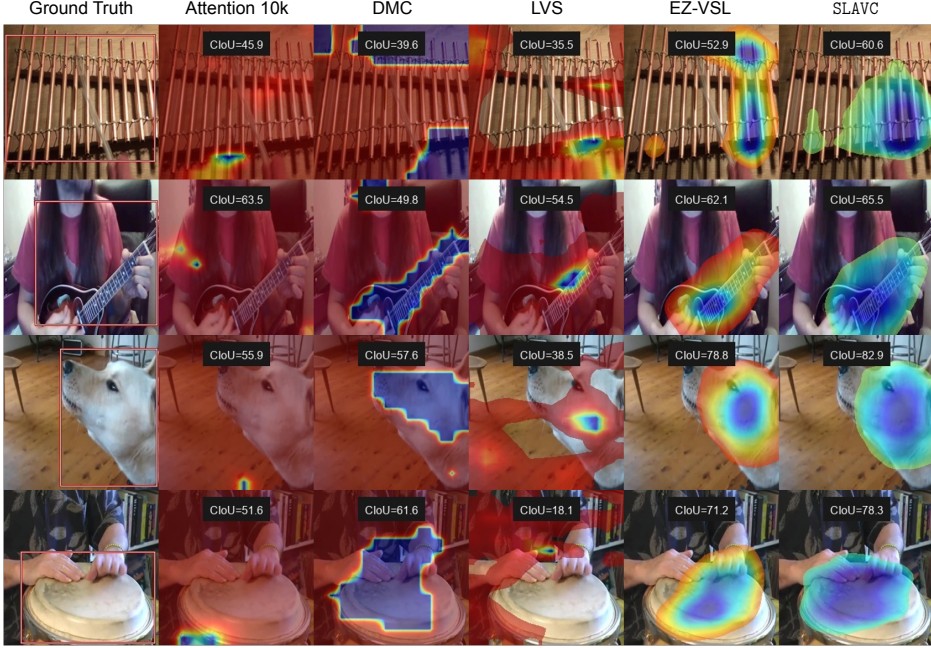

Figure 2: Qualitative results of Attention10k [2], DMC [3], LVS [5], EZ-VSL [7], and the proposed SLAVC on huge sounding objects. Red bounding box and blue maps denote the ground-truth and predictions.

no early stopping for testing heard 110 classes increases the baseline by 5.71%, 5.90%, and 5.22% in terms of AP, max-F1, and LocAcc. When it comes to unheard 110 classes, the proposed SLAVC without OGL outperforms the current state-of-the-art approach by 4.10%, 3.80%, and 3.42% in terms of AP, max-F1, and LocAcc. New state-of-the-art results are achieved in all settings, demonstrating the generalization of our approach to unheard sounding categories in the training set.

## 6 Qualitative Results

In order to qualitatively demonstrate the effectiveness of the proposed SLAVC, we compare visualize attention maps from existing work and our model on various sizes of sounding objects in VGG-SS test set. The qualitative results of of Attention10k [2], DMC [3], LVS [5], EZ-VSL [7], and the proposed

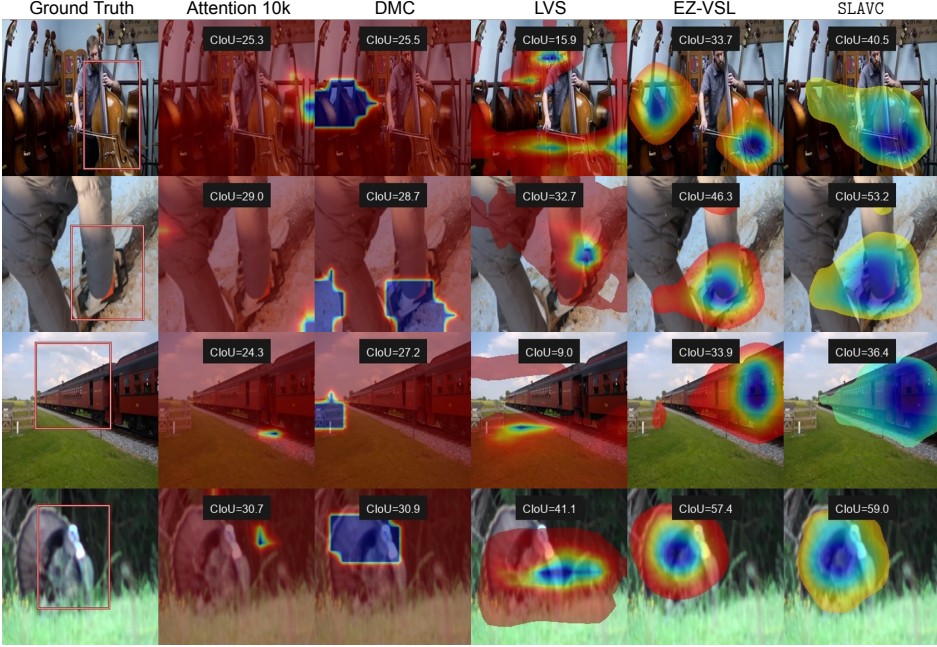

Figure 3: Qualitative results of Attention10k [2], DMC [3], LVS [5], EZ-VSL [7], and the proposed SLAVC on large sounding objects. Red bounding box and blue maps denote the ground-truth and predictions.

SLAVC on huge/large/medium/small sounding objects are shown in Figure 2, 3, 4, and 5. Note that results of our SLAVC are shown on the last column. We can observe that the proposed SLAVC achieves decent localization maps compared to previous approaches in all settings, such as the barking dog on row 3 in Figure 2, the sounding cello on row 1 in Figure 3, and the crying baby on row 1 in Figure 4. For small objects in Figure 5, we also predict better maps for sounding objects than existing work. While our predictions seem to focus more on the actual sources, they are very large compared to the actual objects.

## 7   Broader Impact

This paper seeks to establish a more balanced evaluation protocol for visual sound source localization, that considers both cases with sounding and non-sounding objects. We hope the proposed metrics will lead to more balanced localization algorithms in the future. Improving sound source localization methods can lead to interesting applications, for example, interfacing with blind or low-vision individuals as they navigate the world. However, our work still relies on data collected from internet sources, and thus is likely to hold biases that have not been identified. While we believe current datasets and evaluation protocols are valuable for the development of sound source localization procedure, better curation of the datasets against nefarious bias should be conducted before deployment in real world settings. Advances in audio-visual localization can also be leveraged in surveillance applications, which can potentially have a negative societal impact.

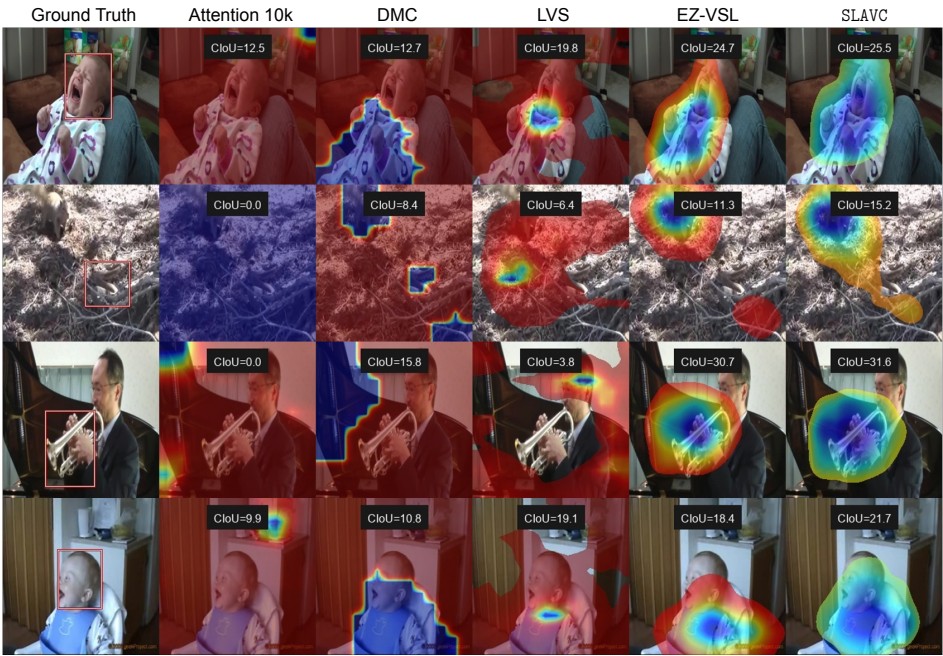

Figure 4: Qualitative results of Attention10k [2], DMC [3], LVS [5], EZ-VSL [7], and the proposed SLAVC on Medium sounding objects. Red bounding box and blue maps denote the ground-truth and predictions.

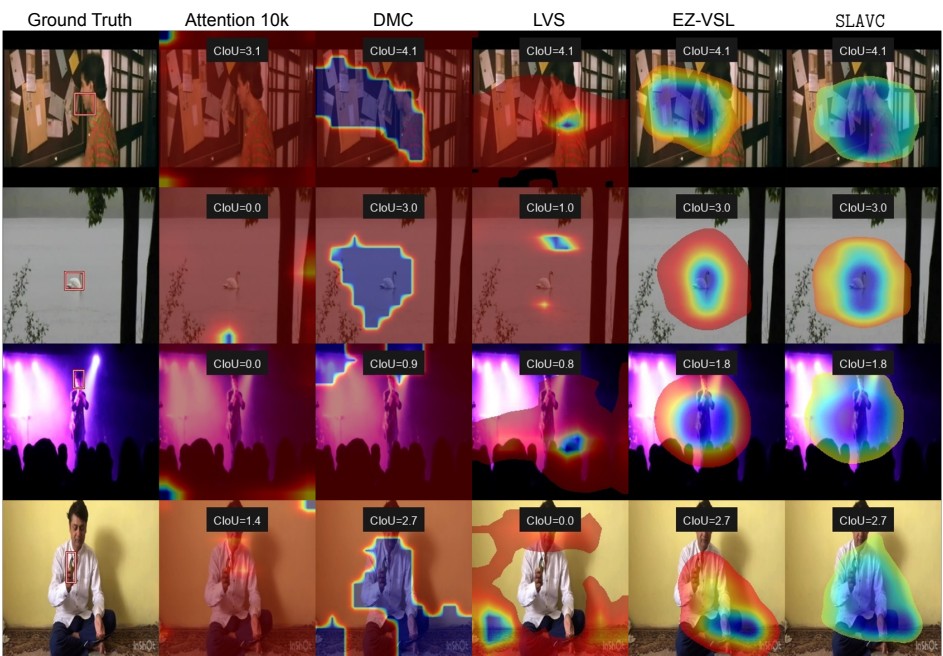

Figure 5: Qualitative results of Attention10k [2], DMC [3], LVS [5], EZ-VSL [7], and the proposed SLAVC on Small sounding objects. Red bounding box and blue maps denote the ground-truth and predictions.