# OpenReview forum: "A Closer Look at Weakly-Supervised Audio-Visual Source Localization"
_NeurIPS.cc/2022/Conference — NeurIPS 2022 Accept_

### Official Review · Reviewer_o43T · 2022-06-14

**Rating:** 7
**Confidence:** 5
**Soundness:** 3 good
**Presentation:** 3 good
**Contribution:** 3 good

**Summary:**

The authors propose a new Momentum-based approach for Visual Sound Localization, termed as MoSVL. It focuses on two key audio-visual sound source localization problems: 1) previous studies are not truly un-supervised and have overfitting problems easily when scaled up to large datasets. 2) the authors claim that all baseline methods fail to consider the silence in the sound localization problem, hence a new mechanism with evaluation protocols is proposed.

**Questions:**

My concerns are elaborate in the above weaknesses field. The lack of comparison and discussions with important related works makes the majority of the paper meaningless.

**Limitations:**

They are discussed in the supplementary material, but it shows up as Section ?? in the checklist. I recommend the authors to pay more attention to the checklist, since this is a very important and useful part in NeurIPS submission.

**Strengths And Weaknesses:**

Strengths:

- [minor +] The audio-visual learning is on the surge and is of great importance. Hence the focused problem has good practical impacts.

- [minor +] The paper is generally easy to follow.

Weaknesses:

- [major -] My biggest concern is that the paper **directly ignores some very very important and relevant works**, DSOL [1] and IEr [2]. They are neither compared nor even discussed/cited in the paper. In particular, DSOL [1] targets to solve the discriminative sounding object localization in multi-sound scenarios **as well as coping with the silent objects**. They suppress those silent but visible objects by multiplying the audio category probability distributions to the visual object distributions, hence those silent objects will not take effect. Besides, DSOL **also propose a manner to effectively evaluate the sound localization performance**. They create a 2x2 grid, with two sounding objects and two silent objects to evaluate cIoU and NSA (Non-Sound Areas). If the model mistakenly localize the sounds to the silent regions, both the cIoU and NSA will perform poorly. In IEr [2], the authors propose to not only deal with the in-the-scene silent objects, but also suppress the off-screen background noise by cross-modal matching. All these two methods are proved to successfully solve the silence problem in the sound localization. However, the authors claim that this is an unsolved problem and it constitutes a very important observation and contribution of this paper.

- [major -] Lack of Novelty. To solve the overfitting problem, the authors propose to add more dropout to the visual features and perform the EMA update on the encoders. However, these are all very basic tricks and couldn't independently serve as a novelty or contribution to the community. Besides, the authors fail to very carefully analyze the effect of dropout or ways to prevent overfitting. Why the dropout works? Why other tricks that help to prevent overfitting won't work, like data augmentation, network parameter normalization, etc.

- [major -] Lack of Demo Video. The lack of demo video weakens the qualitative results of this paper. I could not straight judge the performance when silent objects exist.

- [minor -] No code. As promised by the authors in the checklist, they will include code or data in the supplementary material. Although the authors say that the dataset is too large to submit. The code is not included, which makes doubt the answer [YES] in the checklist and the re-producibility of this paper.

[1] Discriminative Sounding Objects Localization via Self-supervised Audiovisual Matching, Hu et al., NeurIPS 2020.

[2] Visual Sound Localization in-the-Wild by Cross-Modal Interference Erasing, Liu et al., AAAI 2022.

Overall, the ignorance of very relevant works makes the claim, contribution and novelty of the paper poor. The focused problem has already been solved, and the corresponding evaluation protocols have already been proposed, while they are not even discussed in the paper. This makes the main body of the paper meaningless. The left part is the method to solve the overfitting problem, which are quite trivial tricks to me. The authors fail to make their best efforts to delve into the phenomenon and analyze why such methods could alleviate the overfitting problem.

---

> ### Author Response · Authors · 2022-08-02
> **Answer to R4**
>
> **The paper ignores some very important and relevant works, DSOL [1] and IEr [2]. These two methods successfully solve the silence problem in the sound localization. However, the authors claim that this is an unsolved problem.**
>
> Unfortunately, at the time of writing, we were unaware of these two papers, DSOL (NeurIPS'20) and the recently released IEr (AAAI'22). Despite significant differences (which we highlighted below), we agree they are relevant, and should have been cited, discussed and compared with. We added a proper discussion and experimental comparison in the revised manuscript, and hope the reviewer considers this effort. We believe, and support with empirical evidence, that DSOL and IEr do NOT solve the false positive rate problem, as they still tend to overdetect sounding objects when none are visible. We summarize this discussion below.
>
> While both DSOL and IEr propose a method to suppress localization of silent objects, they address a less general problem (ie, with access to more supervision) than ours. Both DSOL and IEr require a clean dataset containing a single source per video, which require a fair amount of human annotation in order to ensure that the training data does NOT contain sound mixtures and that the sound source is on screen. This type of annotation is particularly important to obtain high quality audio/visual prototypes, which in turn are critical for the second learning stage. In contrast, we tackle the problem of visual source localization without assuming knowledge of the number of sources, allowing training to be scaled easily.
>
> As pointed out by the reviewer, DSOL also proposes an evaluation metric (NSA) that measures the ability of a model to ignore visible silent objects. This metric however is computed using an artificially generated 2x2 grid of images, and so it is unclear how it extends to natural images. In contrast, we actually **annotate** real world test samples, where 1) objects are visible but silent, 2) the sound source is active but not visible, or 3) no visible or active sound sources are present. Note that some of these situations are not considered in the NSA metric. We believe the test set we provide will lead to a more realistic and thorough assessment of VSL approaches.
>
> Finally, to compare to our approach, we trained DSOL on the **full VGG-SS** dataset using the code released by the authors (VGG-SS results in the DSOL paper only used a small subset of VGG-SS). We've added the results to Tables 2 and 3 of the paper (see revised manuscript). From Table 2, we see that DSOL still suffers from overfitting issues, with a drop of 3\% if early stopping is not used. Table 3 also shows that DSOL is significantly worse than our method (about 13\% worse precision, 14\% worse max-F1 and 17\% worse AP!). We did not compare on Flickr-SoundNet as this dataset does not have audio label annotations, and thus the audio event model could not be trained. We also do not report results with IEr, as their code has not been released yet.
>
>
> **Lack of Novelty. Dropout and EMA updates are very basic tricks and couldn't independently serve as a novelty or contribution to the community.**
>
> Our paper makes significant advances in both how to learn and how to evaluate unsupervised visual source localization. Simultaneous Instance Discrimination and Localization is a novel contribution of this work with clear benefits, as it demonstrably improves both precision and false positive rates. The extended Flickr-SoundNet and VGG-SS provide a new and effective way to assess a model in a more realistic scenario. We agree that dropout and momentum encoders are basic regularization techniques, but the negative impact of overfitting has clearly been overlooked even after a large number of papers being published on the topic (otherwise, regularization techniques would already be part of prior works). We hope the reviewer takes these contributions into consideration when judging our papers' contributions and novelty.
>
>
> **Why dropout works?**
>
> Regarding visual dropout, it works by encouraging higher spatial redundancy, as the model needs to prevent loss of information when features are dropped (ie it leads to spatially smoother feature maps). This reduces the likelihood of spurious audio-visual alignments.
>
>
> **Demo Video**
>
> We’ve shown several examples of model predictions in Supplementary Material. We’ve also created a demo video. See the following anonymized link:
> https://drive.google.com/file/d/1SDJj1JoZxRIJBcTJL4pKbmcHs3T1DRdm/view
>
>
> **No code**
>
> Unfortunately, we were not able to provide code in supplementary, as we planned. We are however ready to share the code and pre-trained models with the community, as can be seen here: https://anonymous.4open.science/r/MoVSL-1021.

---

> > ### Comment · Reviewer_o43T · 2022-08-04
> > **Response to Authors**
> >
> > I thank the authors for the feedback. Part of my concerns are addressed, e.g., no code and no demo video. Besides, the additional analysis on the novelty generally makes sense.
> >
> > Overall my biggest concern remains that the initial version of the paper ignores DSOL [a] (NeurIPS'20), which makes so many claims in the introduction and related work parts inaccurate. I could understand that IEr (AAAI'22) is recently proposed, so missing this work should not be over-criticized. However, DSOL [a], which targets exactly the same problem of solving silence, ought not to be ignored. Even though I agree with the authors that both DSOL and IEr all require a more demanding single-source dataset for pre-training, the authors could explain the difference between these works' settings. Besides, some of the compared baselines in this work [b] also mention and discuss DSOL. "Unfortunately, at the time of writing, we were unaware of these two papers, DSOL (NeurIPS'20) and the recently released IEr (AAAI'22)." makes me disappointed. The initial rating was mostly derived from the doubt about the authors' efforts in surveying this area, since this is always the first step in doing research.
> >
> > Besides, multiple parts of the claims on the introduction should be revised. For example, in L44 it is incorrect to say that all previous works ignore such problem. The authors are encouraged to highlight the discussions on these works in the introduction, since the title itself contains 'tackle the silence'.
> >
> > Overall, I thank the authors for the rebuttal and your efforts in polishing the manuscript. I will consider raising my score after these revisions.
> >
> > [a] Discriminative Sounding Objects Localization via Self-supervised Audiovisual Matching, Hu et al., NeurIPS 2020.
> >
> > [b] Learning Sound Localization Better From Semantically Similar Samples, Senocak et al., ICASSP 2022.

---

> > > ### Author Response · Authors · 2022-08-04
> > > **Answer to R4's response**
> > >
> > > We are glad that the reviewer took our rebuttal into consideration and appreciate the thoughtful response. Our first revision focused on offering proper experimental comparisons to these works. We have now tried to correct inaccurate statements in the intro and related work. We point the reviewer to lines 45-49 (intro) and 90-101 (related work) of the revised manuscript. We’ve also searched for other places where the claim had to be softened. We quote below the two main additions:
> > >
> > > **Intro (ln45-49)**
> > > >It should be noted that false positive detection is closely related to the silent object detection problem highlighted in recent works [7, 8].
> > > Although these works introduce methods to suppress the localization of silent objects, they require the collection of a clean dataset containing a single source per video. Instead, we aim to tackle this issue without assuming knowledge of the number of sources.
> > >
> > > **Related work (ln90-101)**
> > > >Although understanding when sources are not visible is important to several audio-visual tasks, like in open-domain audio-visual source separation [35, 36], this issue still poses significant challenges. DSOL [ 7] and IEr [ 8] proposed a method to suppress localization of silent objects, and an evaluation metric that measures the ability of a model to ignore visible silent objects. These methods, however, rely on the knowledge of the number of sound sources, not available in most large scale datasets. In contrast, we tackle visual source localization without assuming knowledge of the number of sources. Specifically, we propose a novel Simultaneous Instance Discrimination and Localization framework for weakly-supervised visual source localization with clear benefits, as it demonstrably improves both precision and false positive rates. We also annotate real world test samples that may lead to false positives (beyond silent objects), and propose new metrics for comprehensive evaluation.
> > >
> > > We hope this clarifies how our work stands in relation to DSOL and IEr, and provide these works proper credit for their contributions. Please, let us know if you (or other reviewers) have further suggestions on how to be more precise in addressing this issue.

---

> > > ### Author Response · Authors · 2022-08-07
> > > **Request for additional feedback**
> > >
> > > We’d like to prompt the reviewer whether the later revisions properly addressed the reviewer concerns.
> > >
> > > We sought to clarify in our revision the contributions of DSOL and IEr and how they relate and differ from our work.
> > >
> > > Please let us know if you have further suggestions or concerns.

---

> > > > ### Comment · Reviewer_o43T · 2022-08-08
> > > > **Response to Authors**
> > > >
> > > > I thank the authors for the detailed reply and the corresponding polishments in the draft. I hence increase my score from 2 to 7. Hope to see a better final version of this paper!

---

### Official Review · Reviewer_iTzG · 2022-07-04

**Rating:** 7
**Confidence:** 4
**Soundness:** 4 excellent
**Presentation:** 3 good
**Contribution:** 3 good

**Summary:**

This paper describes two contributions to audio-visual source localization: 1) an evaluation protocol designed to explicitly include negative example cases (where no sound source is visible in the image), and 2) model components designed to address shortcomings in prior work.
The authors compare the proposed model extensions to prior work, using both the proposed evaluation (and extended datasets) and previous evaluation schemes.
Ablation studies quantify the relative contributions of each modeling component across the new metrics.

**Questions:**

I found the description of momentum confusing and lacking context.  There are numerous "dimensions" in which momentum could reasonably be applied here—time, visual extent, training iteration—and a reasonable case could be made for any of them.  (E.g., one should expect smooth variation in the visual extent if sound sources are compact visual objects, and sound activation should be piecewise constant in time for most types.)  I *think* you mean momentum in the training iteration sense, but I'm not 100% certain and could not find any definitive statement to this effect in the text.  Could you please clarify, and explicit about what you're doing?

Similarly, I don't think I fully understood the motivation for SLID.  It seems like the "xid" branch is incentivized to match every visual region to audio from some example in the batch (by virtue of maximizing a softmax, eq 3), but why is this reasonable?  Shouldn't we expect most images to have regions that correspond to no audio at all?

What are the definitions for "small, medium, large, huge" in line 182?

Section 4.1 omits the audio sampling rate, without which the numbers describing the audio encoder are meaningless.  Please clarify.

What is the criterion for determining "best" (bold) in tables 2 and 3?  Of course the "ours" method is bold, but in both tables, there are prior examples (EZ-VSL+OGL) that achieve higher/comparable scores (eg 37.86 vs 37.79).  The authors do state that the experiment is too expensive to support error bars, but some statement of a meaningful difference should be provided.

Several answers to the author checklist are incorrect or left as "N/A".  Specifically:
- 1.c, broken reference (no such text exists)
- 3.d, section 4.1 does not include the specified details
- 4.a, does not appear to cite all tools used (i.e. software), only prior papers describing models
- 4.c, 4.e: answer should be "no", not "N/A" (you are using existing assets, and did not address these points)


**Limitations:**

Some technical limitations are discussed, but no societal implications are mentioned.  The problem being studied here has obvious applications in surveillance that could be mentioned.

**Strengths And Weaknesses:**

Strengths:

The explicit inclusion of negative examples for this task makes a great deal of intuitive sense, and addresses a real shortcoming of previous work in this area.

The evaluation appears to be carefully done, and the proposed modeling components do seem provide a substantive improvement for localization.



Weaknesses:

The proposed model components (section 3.2) could be explained better.
In most cases, I do not see a direct connection between a specific problem with prior models and how these ideas should be expected to help.

---

> ### Author Response · Authors · 2022-08-02
> **Answer to R3**
>
> **Model components could be explained better. In most cases, I do not see a direct connection between a specific problem and how these ideas should be expected to help.**
>
> We summarize below the motivations of each component and how they can help with the issues identified in the paper.
>
> First, to combat overfitting, we adopt 1) visual dropout and 2) slow-moving momentum encoders. Visual dropout is of course a common regularizer. An interesting insight is that to properly address overfitting unusually high visual dropout rates are required. We describe a potential reason for this in lines 133-135. As for the momentum encoder, it prevents overfitting as they provide more stable targets. As mentioned in Ln 140-141: "Since updates to the audio-visual encoders are slowly incorporated into the momentum encoders, the target representations display a smoother behavior during the training process."
>
> To reduce false positives, we explicitly force the model to perform audio-visual instance discrimination in addition to localization. Note that prior works only ask the model to perform instance discrimination through the loss function, and not as a part of the model architecture itself. In our case, the localization module highlights which image regions are most associated with a particular audio, while the instance discrimination module downplays regions that can be better described by the audio of other samples. The key insight is that the model can prevent false detections more effectively, when selecting regions that are simultaneously highlighted in both terms.
>
>
> **There are numerous "dimensions" in which momentum could reasonably be applied here—time, visual extent, training iteration. Which one do we use?**
>
> We use momentum encoders, as commonly used in self-supervised frameworks like MoCo [34]. This means using momentum (over training iterations) to update the encoders used for the target representations (see footnote 3 in Ln139).
>
> We also find the idea of using momentum along other "dimensions" like time or space to be interesting directions for potential future improvements. It is however beyond the scope of this work.
>
>
> **SLID motivation. Shouldn't most images have regions that correspond to no audio at all? Why incentivize all image regions to be matched to a particular audio (through the “xid” branch)?**
>
> We agree that most images have regions that correspond to no audio at all. However, it is NOT true that the "xid" branch requires every visual region to be matched to the corresponding audio. Note that the loss of eq. (4) only operates on the spatial location that is most aligned with the audio (not all locations). That means that, even if some regions are not associated to any audio (eg the softmax outputs a uniform distribution over all audios in the batch), as long as the representations of some visual regions do get aligned with the corresponding audio, the loss can still be minimized.
>
>
> **What are the definitions for "small, medium, large, huge"?**
>
> These categories refer to the different sizes of the ground-truth bounding boxes *as measured by the pixel area*. We clarified the definition of these groups in the revised manuscript. The cutoff between categories is defined in Table 1.
>
>
> **What is the audio sampling rate?**
>
> The audio sampling rate is 22050Hz. We've added to the paper.
>
>
> **What is the criterion for determining "best" (bold) in tables 2 and 3?**
>
> We bolded the best results with and without OGL. OGL is an object prior that can be added to any audio-visual localization method for improved performance. As such, we believe that comparing the pure audio-visual performance (without OGL) is still important. We clarified in the paper.
>
>
> **Several answers to the author checklist are incorrect or left as "N/A"**
>
> We apologize for the incorrectly filled form. We have fixed it.
>
>
> **Societal implications are not mentioned. The problem being studied here has obvious applications in surveillance that could be mentioned.**
>
> We mention that the fairness of current models heavily depend on the biases of the datasets in use, and that a more thorough curation of the datasets against nefarious bias should be conducted before deployment in real world settings. We added a sentence regarding surveillance applications.

---

### Official Review · Reviewer_7NMN · 2022-07-11

**Rating:** 7
**Confidence:** 4
**Soundness:** 3 good
**Presentation:** 3 good
**Contribution:** 3 good

**Summary:**

This paper proposes new metrics and training recipes in order to avoid problematic behavior of sound localization methods which tend to overfit to predicting correctly masks for sounding objects which appear on-screen but completely neglect input images with not on-screen sounding objects. The authors also propose a momentum audio-visual approach which shows state-of-the-art results on widely used benchmark datasets.


**Questions:**

- How can the proposed method be extended towards removing the dependencies on object detection models and object saliency maps?
- The authors can remove the gray background in Figures 2 and 3.

**Limitations:**

In my opinion, the authors aptly address several limitations of their work in the main text (see Section 4.5).


**Strengths And Weaknesses:**

***Pros:***
- The approach is simple and well described so researchers can reproduce the results.
- The paper discusses a really important problem of rejecting regions with not on-screen sounding objects which is beneficial for the community. The paper also extends benchmark datasets with well detailed evaluation dataset extensions to include the capability of models to reject off-screen sounds.
- The results on widely used benchmark datasets are robust and the paper is clearly written.

***Cons:***
- The paper addresses the problem of the models in the literature to overfitting, whereas the main problem is that one could also use hard-negatives and negative samples during training directly (e.g. using a loss function that tries to make the predicted audio-visual mask zero for frames which are not aligned with the input audio). A similar problem (videos with no on-screen sounds) has also been addressed in open-domain audio-visual sound source separation [A] using a multi-instance loss and measuring the off-screen power suppression for sounds which do not appear on-screen. It is not so straightforward why the authors did not try to optimize a straightforward loss on the mask of hard negative examples.
- Picking the confidence threshold also depends on the dataset and thus, although there is no strict dependence on supervised data to train a model with the proposed methodology, this approach also needs supervised data in practice to fine-tune the confidence threshold in order to shift the operating point of the model towards a desired one.
- The ReNet weights which are used come from supervised training on image classification tasks (Line 222: “we initialize the visual encoder with ImageNet [39] pre-trained weights”), please correct me if I am wrong. In this case, the authors either need to remove the terminology about “unsupervised” and replace it with partially/semi/less - supervised or they can show results where their method does not need supervised pre-training to work.

[A] Tzinis E, Wisdom S, Jansen A, Hershey S, Remez T, Ellis D, Hershey JR. Into the Wild with AudioScope: Unsupervised Audio-Visual Separation of On-Screen Sounds. In International Conference on Learning Representations 2021.

I am more than happy to increase my score if the authors address my concerns. To be honest, the only reason that in the current state I vote in favor of NOT accepting this great paper is the misuse of the term “unsupervised” when in all the experiments the backbone image encoder was pre-trained on the whole ImageNet (and this is the reason that I used this rating on the soundness axis).

---

> ### Author Response · Authors · 2022-08-02
> **Answer to R2**
>
> **A similar problem has been addressed in open-domain audio-visual sound source separation (AudioScope).**
>
> We appreciate the reference to open-domain audio-visual source separation and added it to related work. Indeed, being able to understand when sound sources are not visible is of interest to many audio-visual tasks.
>
>
> **Why the authors did not try to optimize a straightforward loss on the mask of hard negative examples?**
>
> Several methods already seek to suppress localization for nonaligned audio-visual pairs. For example, EZ-VSL (ECCV22) uses a contrastive objective that reduces the likelihood of localization when the visual frames and the audio belong to different videos. LVS (CVPR21) finds hard negative regions within a frame, and minimizes a loss that suppresses localization within them. We agree that these more straightforward approaches could potentially address the false localization issue. However, as evidenced by the relatively low AP and max-F1 scores in Table 3, they do not fully address the problem.
>
>
> **This approach needs supervised data to fine-tune the confidence threshold in order to select the desired operating point of the model.**
>
> Technically, the proposed evaluation procedure does not select an operating point, as both max-F1 and AP metrics summarize the performance across all operating points. A higher AP score indicates a higher operating curve on average, and a higher max-F1 score indicates a higher max performance. Having said that, this is a good observation, as deploying the model would require us to choose an operating point. Without validating the chosen confidence threshold on a supervised set, we may not be operating at the highest possible performance. However, given the higher max-F1 and AP scores, our method is still likely to perform better than competing methods regardless of the chosen operating point.
>
>
> **ResNet weights are supervised on image classification tasks. Remove the unsupervised terminology or show results without supervised pre-training. This is “the only reason that in the current state I vote in favor of NOT accepting this great paper.”**
>
> We completely agree with this observation, but we felt the need to use the term "unsupervised" to follow the terminology used in prior works (see [1,2,5,6,7]).
>
> In the literature, "unsupervised source localization" refers to the fact that the model is trained to perform *localization without using ground-truth bounding box annotations*. Since the main goal is to learn to localize (and not visual representation), all prior approaches still use ImageNet pre-trained weights (while still referring to the task as unsupervised localization).
>
> We nevertheless agree with the reviewer, and will try to make this clear in the manuscript. We removed the term "unsupervised" from spotlight locations like the title. To link to prior work, we will still introduce the problem as "unsupervised visual source localization" but add the proper context (see for example new footnote 1 on page 4).
>
> **How can the proposed method be extended towards removing the dependencies on object detection models and object saliency maps?**
>
> We agree that, in principle, audio-visual localization does not need to depend on pre-trained object priors, as the model could learn them if necessary. However, these object priors still seem to be highly effective during inference, as most sound sources are objects.
>
> True large-scale training and further technical advances may reduce the benefits of pre-trained object priors. One may also seek to learn object priors in an unsupervised fashion instead. One potential approach would be to learn from video tracking (ie learn to identify which pixels are likely to move together in a video).
>
>
> **Remove the gray background in Figures 2 and 3.**
>
> Thanks for your suggestion. See manuscript.

---

> > ### Comment · Reviewer_7NMN · 2022-08-03
> > **Response to authors 1**
> >
> > I thank the authors for their effort to answer all my answers and most of my criticisms. However, there is still the most important criticism which still has not been fully addressed by the authors. Pre-training a CNN on a fully labeled image dataset with multiple images of the same object with a different background is by **any** means a model that can be used for a downstream **unsupervised** approach. The approach of this paper and all others cited [1,2,5,6,7] are **weakly supervised** since they are using class labels which helps significantly in understanding class-discriminatory features and drive the model towards picking up the most salient pixel-level features for each class in a much easier way than not having those class labels. If the authors still think that this distinction between *unsupervised* and *weakly supervised* is not so important for their paper, I challenge them to rerun their models with their ResNet trained from scratch and see if there is a difference between a true "unsupervised" and a "weakly supervised" approach as the one proposed here.
> >
> > It is really disheartening that the authors acknowledge that the **unsupervised** terminology is actually wrong (quoting from the response above: "We completely agree with this observation... We nevertheless agree with the reviewer, and will try to make this clear in the manuscript.") and then the authors prefer to continue this misuse of the term simply because other works have done the same mistake (quoting from above: "we felt the need to use the term "unsupervised" to follow the terminology used in prior works (see [1,2,5,6,7]).... To link to prior work, we will still introduce the problem as "unsupervised visual source localization" but add the proper context"). The authors should explicitly state that (or a similar sentence): *In multiple previous works have stated this task as unsupervised sound localization but they actually perform weakly supervised methods which are pre-trained on large image-labeled datasets. Thus, our approach is also weakly-supervised since it is based on strong image priors but we hope that we can extend it to unsupervised approaches where our image encoder will not require labels for pre-training.*
> >
> > Our goal as researchers is not to conceal the truth simply because it is more convenient "*to follow the terminology used in prior works*". Our goal is to write papers that help the community go forward and point out mistakes that prior work has, if need be, to reconsider problems and solutions. At some point there is going to be a work which performs **truly unsupervised** sound event localization and the authors will need to explain how their work is different from the previously wrongly claimed *unsupervised* sound event localization papers, I am betting that the authors would be far better off by pointing this difference right now and help the community and themselves in the long term.
> >
> > My suggestion will be a strong rejection for this very good overall paper until the authors remove all wrongly used **unsupervised** terms from the paper and replace them with **weakly supervised**.

---

> > > ### Author Response · Authors · 2022-08-03
> > > **Answer to R2's response 1**
> > >
> > > We appreciate the prompt and thoughtful response to our rebuttal.
> > >
> > > After reflecting on R2's comments, we completely agreed and decided to address this misconception of prior work more proactively. To this end, we 1) defined the different levels of supervision (unsupervised, weakly supervised, semi-supervised, and fully supervised) that can be used to train VSL models. 2) We explicitly indicated that we are addressing the Weakly Supervised setting, and 3) pointed out that prior works (mistakenly) refer to the problem as unsupervised VSL, as they use supervised pre-trained models. Finally, 4) we edited the rest of the document (including the title) accordingly.
> > >
> > > Quoting from the new revised manuscript (starting at Ln 105):
> > > >**Section 3.1: Preliminaries: Weakly Supervised Visual Source Localization**
> > > >
> > > >Given an audio-visual paired dataset $\mathcal{D}=\{(v_i, a_i): i=1, \ldots, N\}$, visual source localization (VSL) aims at predicting the location of the sources present in sound $a_i$ within the visual frame $v_i$.
> > > >VSL models can be learned from various levels of supervision:
> > > >*Unsupervised VSL* learn to localize without any form of human annotations;
> > > >*Weakly-supervised VSL* forgo bounding-box supervision but may leverage categorical information. This categorical information can be in the form of object labels for visual encoder pre-training, or audio event labels for audio encoder pre-training;
> > > >*Semi-supervised VSL* learns from a small number of bounding-box annotations, together with a large unsupervised or weakly-supervised dataset;
> > > >*Fully supervised VSL* models can learn from large amounts of fully annotated datasets.
> > > >In this paper, we tackle the Weakly-Supervised VSL problem. We note that equivalent prior work, such as [1, 4, 2, 3, 35, 6, 7, 5], often refer to this problem as unsupervised VSL, despite actually addressing the weakly supervised problem, as they use vision models pre-trained on ImageNet for object recognition.
> > >
> > > Please let us know if R2 (or any other reviewers) has further suggestions on how to be more precise in defining or addressing this issue.

---

> > > > ### Comment · Reviewer_7NMN · 2022-08-03
> > > > **Response to authors**
> > > >
> > > > Thanks, your paragraph seems a very good candidate for placing the work of this paper and fixing the issues of the terminology issues by prior work. I will increase my score 2 -> 7 given that the paragraph above goes in the final manuscript. Table 1 still has a wrong unsupervised term in there, please fix and let me know to increase my score.

---

> > > > > ### Author Response · Authors · 2022-08-03
> > > > > **Response to reviewer**
> > > > >
> > > > > Good catch, we missed the caption. It has now been corrected. Thanks again for the thoughtful review!

---

> > > > > > ### Comment · Reviewer_7NMN · 2022-08-06
> > > > > > **As promised**
> > > > > >
> > > > > > As promised, 2->7. Thanks for addressing my comments.

---

### Official Review · Reviewer_2AW4 · 2022-07-12

**Rating:** 8
**Confidence:** 4
**Soundness:** 4 excellent
**Presentation:** 4 excellent
**Contribution:** 3 good

**Summary:**

The authors propose a new method for the self-supervised localization of sounding objects in image. The paper focus on two important limitations of current methodologies 1) they do not handle well sample with no visible or audible sound sources and 2) they are prone to overfitting due to using early stopping criteria on test sets. To address these issues, the main methodological contributions of the paper are - - the introduction of slow-moving momentum audio and visual encoders
- the use of extreme dropout on visual encoders
- the introduction of a loss function that jointly enforces sounding region selection and cross-modal instance discrimination, using different projections for each tasks.

Experiments reveal that the propose method establishes a new state-of-the-art on standard datasets as well as on newly designed datasets that include negative samples. Careful ablation studies on the components of the proposed approach and on the metric parameters are also conducted.

**Questions:**

- In equation (1) and equation (4), the loss function depends on the index i. It should be denoted \mathcal{L}_i to improve clarity. Is it the case that over a mini-batch containing several audio-visual pairs (a_i,v_i), i=1...P, the minimized loss is actually the sum of \mathcal{L}_i for i=1...P ? Moreover, are the sum over j in the loss actually taken over $j\ne i$ ? Please make this precise.

- In eq. (4): typo: I guess the denominator of the second term should contain P_ji instead of P_ij, otherwise it doesn't make sense (the two terms are identical).

- L182: while this is probably obvious to the authors, it is not entirely clear to me what the categories "small, medium, large and huge" refer to. The total image size? The sounding object size? How is this defined?

- L187: the bracketed term "(usually set at γ)" is confusing and should probably be removed.

- Section 4.1: Maybe I missed it but I do not see how the (x,y) locations are selected inside each image to compute local visual features. At random? On a regular grid? And from how many pixel are the visual features are computed?


**Limitations:**

The authors have correctly addressed the limitations of their work, mostly the poor performances on small objects.

**Strengths And Weaknesses:**

## Strengths
- The paper is clear and pleasing to read. The formalization of the method and chosen notations are sound and easy to follow.
- Beyond establishing a new state of the art, it is also a welcome contribution to research reproducibility in this field.
- Experiments are convincing and are rigorously designed to avoid biases, in particular in the choice of metric threshold.

## Weaknesses
- **There is unfortunately a potential fatal mistake in the paper: the precision results of Table 3 seem to have been copy-pasted from Table 2. It is not clear whether the discussions and conclusions drawn by the authors throughout the paper will still hold after fixing this. This must be discussed during the rebuttal phase. Hence, I give the paper a grade of 5, which would become an 8 if this issue is satisfyingly fixed.**
- There are several points in the paper that deserve to be clarified, see next section.

---

> ### Author Response · Authors · 2022-08-02
> **Answer to R1**
>
> **Mistake: Precision in Table 3 copy-pasted from Table 2.**
>
> This is not a mistake. The two metrics (columns "LATEST" in Table 2 and "Precision" in Table 3) are indeed the same. They both represent the precision using the latest checkpoint. So, all the discussion and conclusions of the paper are indeed valid.
>
> Note that, in Table 3, both Flickr and VGG-SS test sets are extended by introducing only negative samples. Thus, this extension of the test sets does NOT affect the computation of Precision (defined as the portion of *actual* sound sources that are correctly localized). In fact, this is why we believe that solely relying on Precision (often called CIoU in the literature) to assess performance is not enough, thus motivating our broad comparative study using metrics that also assess the ability to reject samples without visible sounding sources.
>
> **In equation (1) and equation (4), the loss function should be denoted $\mathcal{L}_i$ to improve clarity.**
>
> Thanks for the great suggestion. We made the change in the revised manuscript.
>
> **Is the final loss the sum of sample losses over a mini-batch?**
>
> The model is trained on the *average* per sample loss: $\mathcal{L}=\frac{1}{P}\sum_{i=1}^P\mathcal{L}_i$. We clarified in the paper.
>
>
> **In eqs (1) and (4), is the sum over j taken over $j \neq i$?**
>
> No, we use the InfoNCE loss [A], where the positive score also takes part in the sum in the denominator. We will use $\sum_{j=1}^P$ instead of $\sum_j$ to make this clear.
>
> [A] Oord, Aaron van den, Yazhe Li, and Oriol Vinyals. "Representation learning with contrastive predictive coding." arXiv:1807.03748, 2018.
>
>
> **Typo in eq. (4): $P_{ji}$ instead of $P_{ij}$.**
>
> Thanks for spotting the typo. We have updated it in the revision.
>
> **How are the categories "small, medium, large and huge" defined?**
>
> These categories refer to the different sizes of the ground-truth bounding boxes *as measured by the pixel area*. We clarified the definition of these groups in the revised manuscript. The cutoff between categories is defined in Table 1.
>
> **Term "(usually set at $\gamma$)" is confusing.**
>
> This is a typo. We meant to say "(usually set at $\gamma=0.5$)".
>
> **How the (x,y) locations are selected inside each image to compute local visual features?**
>
> To compute localized features, we follow prior work like LVS and EZ-VSL, and use the output feature map from a fully convolutional backbone (ResNet-18 w/o the final average pooling and classifier). Thus, the (x,y) locations are indeed selected on a regular grid and are computed from the full receptive field of the last convolutional layer.

---

> > ### Comment · Reviewer_2AW4 · 2022-08-03
> > **Thanks for the clarification**
> >
> > I thank the authors for the clarification, I had indeed missed that the precision would not be affected by the addition of negative samples. I would recommend to make an explicit note on this in the paper, as I believe other people than me could get confused as well.
> >
> > As promised, I hence change my overall score from 5 to 8.

---

### Author Response · Authors · 2022-08-02
**Comment to all reviewers**

We thank all reviewers for the insightful feedback. We believe that it already improved the overall quality of our paper. We will address each reviewer in a separate thread. When appropriate, we made changes to the paper directly. See the revised manuscript for reference (all changes in the manuscript are shown in red). We encourage the reviewers to reach out during the discussion period, if you have any remaining questions.

---

### Author Response · Authors · 2022-08-09
**Closing remarks**

We sincerely thank all reviewers for the thoughtful responses and constructive feedback. We truly believe they improved the quality of the paper overall.

Weakly-supervised audio-visual source localization is a challenging task that aims to predict the location of visual sound sources for enhanced audio-visual perception, while relying on minimal human annotation.
This is critical for many applications in the community, such as robotic perception and egocentric video understanding. We thus believe that advances in visual source localization will have many downstream benefits.

Our paper makes significant advances in both how to learn and how to evaluate weakly-supervised visual source localization.
We propose “Simultaneous Instance Discrimination and Localization” - a novel contribution of this work with clear benefits, as it demonstrably improves both precision and false positive rates.
We also extended Flickr-SoundNet and VGG-SS to address a critical limitation of these established benchamark datasets, and provide a new and effective way to assess a model in a more realistic scenario.

Reviewers played an important role in improving our paper. It was pointed out the misuse of the term “unsupervised” and the existence of relevant works on identifying silent objects.
In rebuttal, we addressed these concerns. To clarify the different levels of supervision that can be used to train VSL model, we formally defined unsupervised, weakly supervised, semi-supervised, and fully supervised VSL.
We also explicitly indicated that we (as well as other prior work) are addressing the Weakly Supervised setting.
Regarding prior work on silent objects, we clarified how our work stands in relation to DSOL (NeurIPS’2020) and IEr (AAAI’2022), and gave them proper credit for their contributions. We also added DSOL to the list of methods that we benchmarked using the new evaluation protocol.

We appreciate that all of the reviewers recognized our contributions and believe that our study has the potential to inspire more audiences attending NeurIPS.

---

### Meta-Review · Area_Chair_j7Tx · 2022-08-23

**Recommendation:** Accept
**Confidence:** Certain

**Metareview:**

The authors seem to have addressed most if not all of the reviewers recommendations, leading to a much improved paper compared to the initial manuscript. The updated scores from the reviewers reflect the major improvements and therefore I recommend this paper be accepted in its updated form.

**Award:**

No

---

### Decision · Program_Chairs · 2022-09-14

Accept